# Salt Tolerance Indicators in 'Tahiti' Acid Lime Grafted on 13 Rootstocks

**Gabriel O. Martins** [1] , **Stefane S. Santos** [2], **Edclecio R. Esteves** [2], **Raimundo R. de Melo Neto** [2],
**Raimundo R. Gomes Filho** [1], **Alberto S. de Melo** [3] , **Pedro D. Fernandes** [4], **Hans R. Gheyi** [4] ,
**Walter S. Soares Filho** [5] **and Marcos E. B. Brito** [2,*]

[1] Post Graduate Program in Water Resources, Universidade Federal de Sergipe (UFS), Campus of São Cristóvão, São Cristóvão 49100-000, SE, Brazil

[2] Center of Agrarian Sciences of Sertão (CCAS), Universidade Federal de Sergipe (UFS), Campus of Sertão, Nossa Senhora da Glória 49680-000, SE, Brazil

[3] Center of Biological and Health Sciences, Universidade Estadual da Paraíba (UEPB), Campus I, Campina Grande 58429-500, PB, Brazil

[4] Center of Tecnologia and Natural Resources (CTRN), Universidade Federal de Campina Grande (UFCG), Campina Grande 58429-900, PB, Brazil

[5] National Research Center of Cassava and Fruit Crops (CNPMF), Empresa Brasileira de Pesquisa Agropecuaria, Cruz das Almas 44380-000, BA, Brazil

\* Correspondence: marcoseric@academico.ufs.br

**Abstract:** The citrus yield is limited by soil and/or water salinity, but appropriate rootstocks can ensure the sustainability of the production system. Therefore, the objective of the present research was to evaluate the salt content in the soil and the production and physiological aspects of the 'Tahiti' acid lime combined with thirteen rootstocks, irrigated with saline water in the first two production years to identify indicators of salt tolerance. The rootstocks evaluated were: 'Santa Cruz Rangpur' lime, 'Indio', 'Riverside' and 'San Diego' citrandarins, 'Sunki Tropical' mandarin, and eight hybrids, obtained from the Citrus Breeding Program of Embrapa Cassava and Fruits. The waters used had three saline levels: 0.14, 2.40, and 4.80 dS m$^{-1}$, in a randomized block adopting a split-plot design, with rootstocks in the plots and saline waters in the subplots, with four replicates. From August 2019 to February 2021, fruit harvests and agronomic traits were measured. At the end of each production year, the soil characteristics, leaf gas exchange, and chlorophyll *a* fluorescence analysis were performed. It was concluded that: (1) the effects of water salinity on citrus are of osmotic nature, reducing gas exchange, (2) the salinity did not significantly damage the photosynthetic apparatus until the second year of production, and (3) using more stable, salt-tolerant rootstocks makes it possible to cultivate 'Tahiti' acid lime under irrigation with waters of 2.4 dS m$^{-1}$ electrical conductivity.

**Keywords:** chlorophyll *a* fluorescence; *Citrus* spp.; fruit production; *Poncirus* hybrids; salt balance

---

## 1. Introduction

Brazilian citrus is composed of several species, especially those belonging to the genus *Citrus*, which is formed, among other species, by sweet oranges (*C.* × *sinensis* (L.) Osbeck) [1] and Tahiti acid lime (*C.* × *latifolia* (Yu. Tanaka) Tanaka). Regarding 'Tahiti' acid lime, known as 'Tahiti' lemon, it stands out for its increase in terms of both harvested area (58,438 ha) and production, with a total of 1,585,215 tons of fruits, highlighting the states of São Paulo, Pará, Minas Gerais, Bahia, Rio de Janeiro, Ceará, Paraná, Espírito Santo, Rio Grande do Sul, and Sergipe, the last-mentioned at the tenth position on the national scene [2].

'Tahiti' acid lime production is very important in Brazil, not only from an economic point of view but also socially, with the generation of employment and income, because it allows families to remain in the field, with work and dignity [3]. Despite its socio-economic importance, its cultivation in the Northeast region faces some adversities, which lead to a low yield, about 11.9 t ha$^{-1}$ [2], considering its potential of 40 t ha$^{-1}$ [4].

The yield of citrus plants in northeastern Brazil may be optimized with the use of irrigation [5]. However, the waters available in this region of the country, mainly groundwaters, have high salt contents [6], which can cause problems since citrus plants are considered sensitive to salinity [7,8], which reduces their production capacity as their threshold salinity is around 1.4 dS m$^{-1}$ in saturation extract of soil and 1.1 dS m$^{-1}$ in irrigation water [6].

However, this response may vary depending on the scion/rootstock combination used and the management of the production system [9,10], which denotes the importance of identifying rootstocks that can confer tolerance to the scion variety, to obtain economically viable yields, even under saline conditions.

Thus, the screening of rootstocks may make it possible to use saline waters and soils, as they enable the sustainability of citrus cultivation in areas subject to this abiotic stress, common in northeastern Brazil, which is a large citrus-producing center. It is important to choose a rootstock that can give the scion desirable agronomic characteristics, such as early fruit production, low height, resistance to pests and diseases, and tolerance to abiotic stresses [11]. Few studies have been conducted on scion/rootstock combinations with salt tolerance for Brazilian citrus production [12].

Regarding tolerance to salt stress, researchers have studied the effects of salinity on nutritional imbalance and ionic interactions in plant tissue [8–10,13]. In glycophytes, this tolerance/adaptation to salt stress can be observed even under small accumulation of sodium (Na$^+$) and chloride (Cl$^-$) in the aerial parts or in the plant as a whole, a process that is related to the ability to exclude ions, especially in the root system [13], which makes the rootstock an essential component in the formation of the citrus plant.

Thus, the objective of this work was to study the salt accumulation in the soil, in addition to the production and physiological aspects of combinations of citrus rootstocks with 'Tahiti' acid lime irrigated with saline water during the first two years of cultivation, aiming to identify the indicators of salt tolerance.

## 2. Materials and Methods

The experiment was carried out at the Experimental Farm of the Embrapa semi-arid region, located in the municipality of Nossa Senhora da Glória, Sergipe, Brazil (10°12′18″ S, 37°19′39″ W, and 294 m altitude). Using the spreadsheet of water balance integration with climate classification proposed by Sousa and Brito [14], it is possible to observe the 'As' climate classification, relative to tropical climate. The rainy season is between April and August, with a concentration in May, June, and July. The region has low relative air humidity and wide thermal variation between day (between 28 and 35 °C) and night (between 18 and 21 °C).

The accumulated precipitation in the 26 months of the study was 1387.4 mm, being 554.3 mm in 2019, 750.9 mm in 2020, and 82.2 mm in January and February 2021, values that are within the average for the region, but below the water requirements of most citrus species, within the range from 900 to 1500 mm annually [15].

The experimental design was randomized blocks, with 13 scion/rootstock combinations under 3 levels of saline water, using the split-plot scheme with 4 replicates, as follows:

(a)   Plot: 13 scion/rootstock combinations (genotypes), with 'Tahiti' acid lime grafted onto 13 rootstocks (Table 1), all from the Citrus Breeding Program of Embrapa Cassava and Fruits—CBP.

(b)   Subplot: Three types of water (salinities), with electrical conductivities (ECw) of 0.14, 2.4, and 4.8 dS m$^{-1}$, with the first corresponding to water from the São Francisco River and the other two obtained by diluting tube well water. The chemical characteristics are presented in Table 2, with the water from the São Francisco River, until reaching the desired EC levels, with values measured using a portable microprocessor conductivity meter with automatic temperature adjustment at 25 °C.

**Table 1.** Rootstocks (genotypes) studied under water salinity when grafted on 'Tahiti' acid lime (*Citrus × latifolia* (Yu. Tanaka) Tanaka).

| | Rootstock | Origin |
|---|---|---|
| 1. | 'Rangpur Santa Cruz' lime | *Citrus × limonia* Osbeck |
| 2. | 'Indio' citrandarin | *C. sunki* (Hayata) hort. ex Tanaka × *Poncirus trifoliata* (L.) Raf. |
| 3. | 'Riverside' citrandarin | *C. sunki × P. trifoliata* |
| 4. | 'San Diego' citrandarin | *C. sunki × P. trifoliata* |
| 5. | 'Sunki Tropical' mandarin | *C. sunki* |
| 6. | TSKC × TRBK—007 | *C. sunki × P. trifoliata* |
| 7. | TSKFL × TRBK—030 | *C. sunki × P. trifoliata* |
| 8. | TSKC × CTTR—012 | *C. sunki × [C. × sinensis* (L.) Osbeck × *P. trifoliata*] |
| 9. | TSKFL × CTTR—013 | *C. sunki × (C. × sinensis × P. trifoliata)* |
| 10. | HTR—069 [1] | *C. × sinensis × (C. × sinensis × P. trifoliata)* |
| 11. | TSKC × (LCR × TR)—040 [2] | *C. sunki × (Citrus × limonia × P. trifoliata)* |
| 12. | TSKC × (LCR × TR)—059 [3] | *C. sunki × (Citrus × limonia × P. trifoliata)* |
| 13. | TSKC × CTARG—019 | *C. sunki × (C. × sinensis × P. trifoliata)* |

TSKC = common 'Sunki' mandarin; TRBK = P. trifoliata 'Benecke'; TSKFL = 'Sunki of Florida' mandarin; CTTR = 'Troyer' citrange; HTR - 069 = trifoliate hybrid, called citrangor due crossing of 'Pera' sweet orange with 'Yuma' citrange; LCR = 'Rangpur' lime; TR = P. trifoliata; CTARG = 'Argentina' citrange. [1] Citrangor in the registration process as a rootstock variety, by Embrapa, in the National Cultivar Register (RNC) of the Brazilian Agriculture, Livestock and Supply Ministry (MAPA), called 'BRS Santana'. [2] Citrimoniandarin registered as a rootstock variety, by Embrapa, in the RNC-MAPA under the name 'BRS Tabuleiro'. [3] Citrimoniandarin registered as a rootstock variety, by Embrapa, in the RNC-MAPA under the name 'BRS Bravo'.

**Table 2.** Chemical characteristics of the water from the tube well used in the preparation of water of 2.4 and 4.8 dS m$^{-1}$.

| EC | pH | Ca$^{2+}$ | Mg$^2$ | Na$^+$ | K$^+$ | CO$_3{}^{2-}$ | HCO$_3{}^-$ | SO$_4{}^{2-}$ | Cl$^-$ |
|---|---|---|---|---|---|---|---|---|---|
| dS m$^{-1}$ | | | | | mmol$_c$ dm$^{-3}$ | | | | |
| 30.80 | 7.20 | 30.80 | 78.88 | 148.22 | 2.35 | 0.00 | 7.36 | 3.37 | 289.0 |

EC = electrical conductivity at 25 °C.

The experimental unit consisted of one plant per pot, and the application of waters with different salinity levels began 30 days after transplanting (DAT) of the seedlings in pots adapted as lysimeters, which continued throughout the evaluation period along with the soil water balance.

The plants of nucellar origin, identified based on leaf morphological characteristics, with good formation and representative of each rootstock, were grafted on the acid lime clone 'Tahiti CNPMF-01'. The seedlings were produced by the Tamafe® Nursery, following the certified seedlings process. These were produced in plastic bags with a capacity of 2 L, filled with the commercial substrate (Basa-plant®). The seedlings remained under this condition until they were suitable to be transplanted, and the period before transplanting was approximately 300 days.

The seedlings were taken to the experimental farm of the Embrapa semi-arid region, where they were transplanted into pots adopted as lysimeters with a capacity of 60 L. The lysimeters were filled with sieved (10 mesh) Ultisol from the nearby area with the addition of organic manure.

Until 30 DAT, the plants received water of low electrical conductivity (ECw), coming from the local supply system; after this period, water of different types according to treatments was applied. Irrigations were performed with a drip irrigation system every two days.

Irrigation management was carried out based on the water balance method to replenish daily mean water consumption by the plants, plus a leaching fraction (LF) of 0.10, to avoid excessive accumulation of salts in the root zone, using Equation (1) to calculate the volume. The drained water was collected by a hose connected to the base of each lysimeter.

$$Vi = \frac{(Va - Vd)}{1 - LF} \tag{1}$$

where: Vi = volume to be applied in the irrigation event (mL), Va = volume applied in the previous irrigation event (mL), Vd = volume of water drained (mL), and LF = leaching fraction (10% = 0.10).

The nutritional management of plants was based on soil analysis and followed the recommendations presented in [15]. In the first year of production, in fertilization (N, $P_2O_5$, and $K_2O$), at weekly intervals, fertigation was performed applying 4.27 g plant$^{-1}$ of urea, 7.69 g plant$^{-1}$ of purified monoammonium phosphate (MAP), and 1.28 g plant$^{-1}$ of KCl.

In the second year of production, after a new soil analysis, weekly fertilization was adjusted to 9.40 g plant$^{-1}$ of urea, 5.12 g plant$^{-1}$ of purified MAP, and 3.84 g plant$^{-1}$ of KCl. In both years of production, the plants received all the micronutrients necessary for their development, also via irrigation water, as recommended in [15].

Other tillage practices related to the control of weeds and pests were adopted, particularly manual removal of invasive plants that appeared in the pots and application of pesticides when pests occurred [15]. Additionally, cleaning operations in the irrigation system, preparation of the irrigation water, formative pruning of the plants, mowing in the inter-rows of the pots, as well as cleaning and periodic maintenance of lysimeters were carried out when necessary.

At the end of each production year, soil samples (0–0.20 m) were collected in each experimental plot to obtain the saturation extract, to determine EC, pH, and the contents of $Ca^{2+}$, $K^+$, $Na^+$, and $Mg^{2+}$, to calculate the sodium adsorption ratio (SAR) and estimate the exchangeable sodium percentage (ESP), using the relationship between ESP and SAR reported by Richards [16].

During the reproductive stage, starting around 300 DAT, as the fruits reached the harvest stage [17] they were collected for subsequent counting and weighing to determine the number of fruits per plant (NFPL), the weight of fruits per plant (WFPL) (g plant$^{-1}$), using a scale with a resolution of 0.01 g, and to calculate the average fruit weight (AFW) (g fruit$^1$).

Physiological analyses of the plants were also performed at 270 and 720 DAT in the morning. Chlorophyll *a* fluorescence analysis was determined in the first mature leaf from the stem apex, in good phytosanitary condition and fully expanded, using an Opti Science OS5p pulse-modulated fluorometer, based on the OJIP protocol to determine the quantum efficiency of the photosystem II (PSII) (Fv/Fm). Thus, the fluorescence induction variables were determined: initial fluorescence (F0) and maximum fluorescence (Fm).

These data were then used to calculate the variable fluorescence (Fv, where Fv = Fm − F0) and maximum quantum efficiency of photosystem II, using the Fv/Fm ratio [18]. The analyses were performed after adaptation of the leaves to the dark for 40 min, based on a previous test, using a clip of the device to ensure that all the primary acceptors were oxidized, i.e., with the reaction centers opened.

In the same period of fluorescence determination, gas exchange analysis was performed using an infrared gas analyzer (IRGA) (LCpro+), with constant light of 1200 μmol of photons m$^{-2}$ s$^{-1}$, on the third leaf of the plant counted from the apex, obtaining the following photosynthetic variables: $CO_2$ assimilation rate (A) ($\mu mol_{CO2}$ m$^{-2}$ s$^{-1}$), transpiration (E) (mol $H_2O$ m$^{-2}$ s$^{-1}$), stomatal conductance (gs) (mol $H_2O$ m$^{-2}$ s$^{-1}$), and internal $CO_2$ concentration (Ci) ($\mu mol_{CO2}$ mol$^{-1}$). These data were used to quantify the intrinsic

water use efficiency (WUEi) by the A/E ratio (($\mu$mol m$^{-2}$ s$^{-1}$) (mol H$_2$O m$^{-2}$ s$^{-1}$)$^{-1}$) and the intrinsic carboxylation efficiency ($\Phi$c (CEi)) by the A/Ci ratio [19].

The data obtained were subjected to ANOVA by the F test ($p \leq 0.05$); following this, the production data were analyzed using boxplots from package ggplot2 of the RStudio software. For the significant effect of the water salinity, the Tukey test ($p \leq 0.05$) was employed, and for the genotype factor, the cluster test was carried out (Scott–Knott, $p \leq 0.05$).

The obtained production data were correlated with the soil attributes, using the corrplot package in RStudio$^{\circledR}$.

## 3. Results

### 3.1. Chemical Analysis of Soil

In the first year, there were variations in the electrical conductivity of the saturation extract (ECse), 1.5 to 10.8 dS m$^{-1}$, for pH 5.6 to 6.0, and in SARse between 2.16 and 35.04 (mmol L$^{-1}$)$^{0.5}$, characterizing the soil as non-saline non-sodic (without salinity problems) under irrigation with water of 0.14 dS m$^{-1}$, and those that received waters of 2.4 and 4.8 dS m$^{-1}$ as saline-sodic soil, with the ESP values $\geq$ 15% and ECse $\geq$ 4 dS m$^{-1}$ (Table 3) [20].

**Table 3.** Mean values of soil chemical attributes (0–0.20 m depth) at each salinity level. Samples were taken from the plots in the first and second years of production.

| Water Salinity | Year | pH | ECse | Ca$^{2+}$ | K$^+$ | Na$^+$ | Mg$^{2+}$ | SARse | ESP |
|---|---|---|---|---|---|---|---|---|---|
| dS m$^{-1}$ | | | dS m$^{-1}$ | | mmol$_c$ L$^{-1}$ | | | (mmol L$^{-1}$)$^{0.5}$ | |
| 0.14 | 1 | 5.6 | 1.5 | 1.1 | 7.4 | 2.4 | 1.1 | 2.2 | 1.70 |
| 2.40 | 1 | 5.9 | 5.5 | 0.8 | 13.2 | 36.2 | 1.6 | 34.5 | 34.5 |
| 4.80 | 1 | 6.0 | 10.8 | 9.6 | 15.6 | 81.5 | 2.7 | 35.0 | 35.0 |
| 0.14 | 2 | 5.3 | 2.0 | 2.2 | 13.3 | 1.2 | 2.5 | 0.5 | 0.3 |
| 2.40 | 2 | 4.9 | 6.6 | 6.1 | 11.8 | 21.7 | 24.1 | 5.1 | 5.8 |
| 4.80 | 2 | 4.9 | 9.5 | 12.9 | 9.4 | 32.2 | 38.9 | 6.3 | 7.5 |

pH = hydrogen potential; ECse = electrical conductivity of saturation extract; SARse = sodium adsorption ratio of saturation extract; ESP = exchangeable sodium percentage, estimated based on the relationship between SARse and ESP according to Richards [16].

In the second production year, the ECse, pH, and SARse ranged from 2.0 to 9.5 dS m$^{-1}$, 5.3 and 4.9, and 0.5 and 6.3 (mmol L$^{-1}$)$^{0.5}$, respectively (Table 3). Another difference that draws attention is the reduction of K$^+$ contents in the soil in the second year (13.3 to 9.4 mmol$_c$ dm$^{-3}$).

It was also observed that the Na$^+$/K$^+$ ratio in the saturation extract increased with the increase in the salinity of the applied water, being, in the first year, from 0.32 in soil irrigated with an EC of 0.14 dS m$^{-1}$, to 5.22 in soil under irrigation using waters with an EC of 4.8 dS m$^{-1}$. In the same line, in the second year, the increase of the relation Na$^+$/K$^+$ was from 0.09 to 3.42 in soil under irrigation water of 0.14 and 4.8 dS m$^{-1}$, respectively.

### 3.2. Analysis of Production

The increase in salinity significantly affected the yield of 'Tahiti' acid lime grafted onto the rootstocks in the first year of production (Table 4), with effects of the interaction between rootstocks and water salinity levels on the number of fruits per plant (NFPL), weight of fruits per plant (WFPL), and average fruit weight (AFW) ($p \leq 0.05$). When considering the single factors, significant effects of rootstocks were observed for AFW ($p \leq 0.01$), and salinity affected ($p \leq 0.01$) all variables analyzed.

**Table 4.** Summary of analysis of variance of number of fruits per plant (NFPL), weight of fruits per plant (WFPL) (g plant$^{-1}$), and average fruit weight (AFW) (g fruit$^{-1}$) of combinations of 'Tahiti' acid lime (*Citrus* × *latifolia* (Yu. Tanaka) Tanaka) with 13 rootstocks 270 days after the onset of saline water irrigation.

| Variation Factors | | Mean Squares | | |
|---|---|---|---|---|
| | | NFPL | WFPL | AFW |
| Block | 3 | 58.99 [ns] | 231,500.83 [ns] | 323.33 ** |
| Genotype (Gen) | 12 | 119.92 [ns] | 298,621.30 [ns] | 47.91 ** |
| Error 1 | 36 | 85.84 | 191,893.86 | 40.37 |
| Salinity (Sal) | 2 | 10,281.82 ** | 27,051,122.46 ** | 251.53 ** |
| Gen × Sal | 24 | 175.66 * | 284,288.68 * | 72.519 * |
| Error 2 | 78 | 72.50 | 168,967.88 | 29.59 |
| CV 1 (%) | | 31.71 | 32.96 | 13.97 |
| CV 2 (%) | | 30.52 | 30.93 | 11.96 |
| Mean | | 29.215 | 1329.011 | 45.471 |

[ns] = not significant; * and ** = significant at 0.05 and 0.01 probability levels, respectively; CV = coefficient of variation; DF = degrees of freedom; Genotype = combination of scion/rootstock.

The increased water salinity reduces the number of fruits per plant, but differently among the rootstocks (Figure 1). Under 0.14 dS m$^{-1}$ (Figure 1A), there were no differences between genotypes, according to the means from the cluster test, although genotypes 6 and 7, corresponding to hybrids between the common 'Sunki' mandarin (TSKC) and *P. trifoliata* Benecke (TRBK)—007 (TSKC × TRBK—007) and between the 'Sunki of Florida' mandarin (TSKFL) and the selection of trifoliate orange (*P. trifoliata*) (TSKFL × TRBK—030), respectively, in addition to the citrandarins 'San Diego' (genotype 4) and 'Indio' (genotype 2), had lower variability in the number of fruits (Figure 1).

Water with 2.4 dS m$^{-1}$ (Figure 1B) caused a reduction in the number of fruits per plant, with the distinction of two groups of genotypes, with the lowest means observed in genotypes 8, 9, 11, and 13, corresponding to the hybrids TSKC × 'Troyer' citrange (CTTR)—012, TSKFL × CTTR—013, TSKC × ('Rangpur' lime (LCR) × *P. trifoliata* (TR))—040, and citrimoniandarin and TSKC × 'Argentina' citrange (CTARG)—019, respectively, which denotes higher sensitivity, already at this salinity level. On the other hand, citrandarins TSKC × TRBK—007, 'San Diego' and TSKFL × TRBK—030, and the citrimoniandarin TSKC × (LCR × TR)—059, besides being in the group of genotypes with the highest number of fruits, showed greater stability in the first year.

Irrigation with waters of 4.8 dS m$^{-1}$ (Figure 1C) did not show distinction among genotypes according to the Scott–Knott test; however, in genotypes such as 'Rangpur Santa Cruz' lime, TSKFL × TRBK—030, TSKC × (LCR × TR)—059, and TSKC × CTARG—019, maximum reductions were observed in the number of fruits per plant compared to the values obtained when the plants were irrigated with waters of 0.14 dS m$^{-1}$. Furthermore, under the condition of higher water salinity, the mean values observed mostly showed less variation, which can be observed by the size of the boxplot.

Weight of fruits per plant (Figure 2A–C) was reduced by salinity in all scion/rootstock combinations; however, distinct groups were formed only when applying water with lower salinity, with the highest values of fruit weight in plants grafted with 'Rangpur Santa Cruz' lime, 'Indio' and 'Riverside' citrandarins, 'Sunki Tropical' mandarin, and with hybrids TSKFL × TRBK—030, HTR—069, and TSKC × (LCR × TR)—059, according to the Scott–Knott test ($p \leq 0.05$).

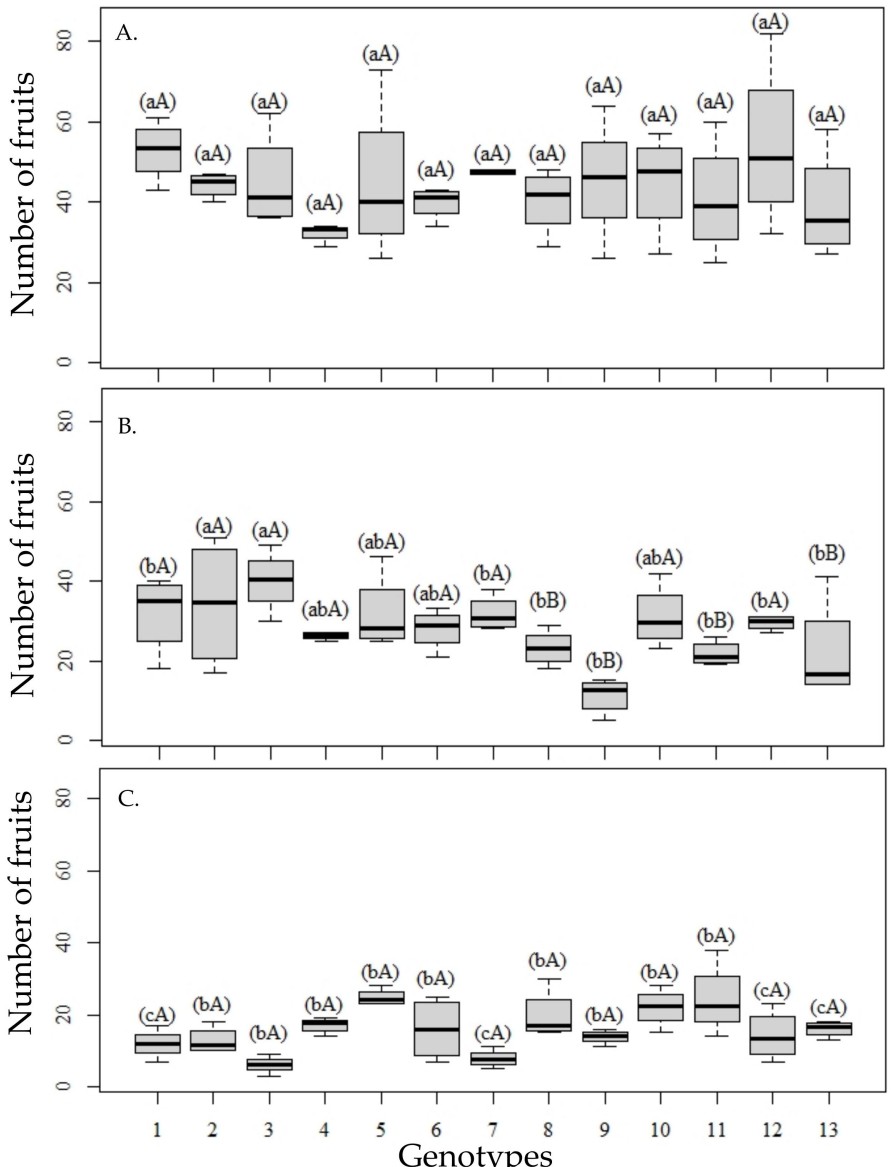

**Figure 1.** Boxplot relative to the mean number of fruits per plant of 13 citrus rootstocks in combination with 'Tahiti' acid lime (*Citrus × latifolia* (Yu. Tanaka) Tanaka) irrigated with water of 0.14 (**A**), 2.40 (**B**), and 4.80 dS m$^{-1}$ (**C**) in the first year. For identification of genotypes, see Table 1. Boxplots with the same lowercase letter do not differ statistically, according to the Tukey test between salinity levels ($p \leq 0.05$), and those with the same uppercase letter belong to the same genotype group, according to the Scott–Knott test ($p \leq 0.05$).

When analyzing the effect of salinity on the mean fruit production per rootstock plant, it is possible to verify greater relative reductions in plants grafted with 'Santa Cruz Rangpur' lime, 'Indio' citrandarin, and the hybrid TSKFL × TRBK—030, with a reduction in fruit weight greater than 60% when the plants were irrigated with water of 4.8 dS m$^{-1}$ (Figure 2C) compared to the results obtained in these combinations irrigated with water of 0.14 dS m$^{-1}$ (Figure 2A).

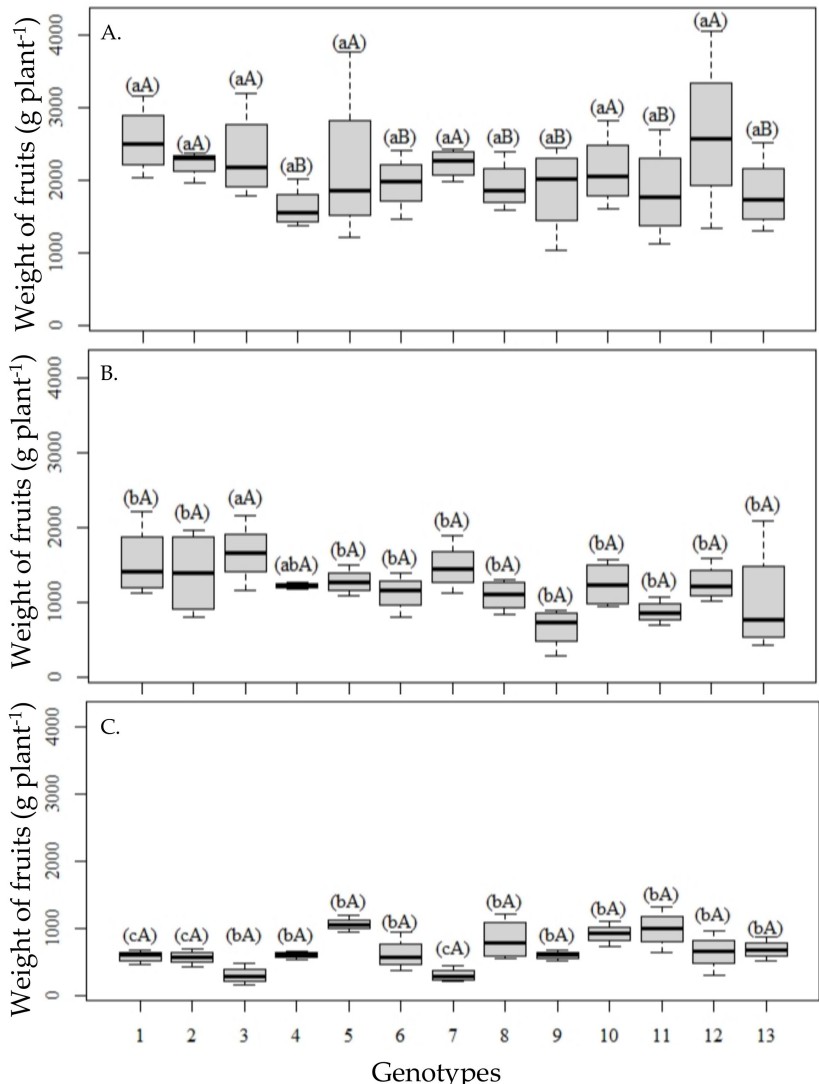

**Figure 2.** Boxplot relative to the average weight of fruits per plant of 13 citrus rootstocks in combination with 'Tahiti' acid lime (*Citrus × latifolia* (Yu. Tanaka) Tanaka) under irrigation with waters of 0.14 (**A**), 2.40 (**B**) and 4.80 dS m$^{-1}$ (**C**), in the first year. For identification of genotypes, see Table 1. Boxplots with the same lowercase letter do not differ statistically, according to the Tukey test between salinity levels ($p \leq 0.05$), and those with the same uppercase letter belong to the same genotype group, according to the Scott–Knott test ($p \leq 0.05$).

The 'Sunki Tropical' mandarin, HTR—069 citrangor, and TSKC × (LCR × TR)—059 citrimoniandarin, on the other hand, gave the 'Tahiti' acid lime greater stability in variable weight of fruits (Figure 2), even with the increase in the salinity level in the first year of cultivation, i.e., there was a smaller reduction in the mean production of fruits with the increase in water salinity.

The salinity caused a loss of production in the number and weight of fruits per plant, but in general, the plants tried to maintain the mean weight of fruit in the first year of cultivation (Figure 3A–C), even with the increase in water salinity, with a significant reduction in the mean weight of 'Tahiti' fruit when grafted on 'Riverside' and 'San Diego' citrandarins.

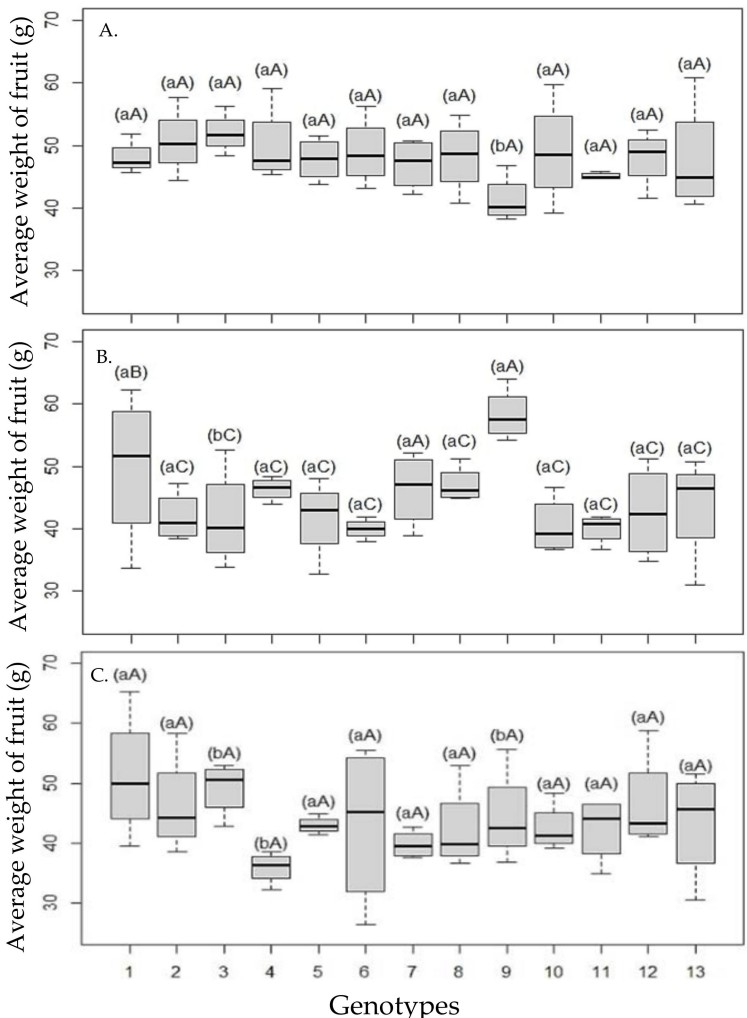

**Figure 3.** Boxplot relative to the average weight of fruit of 13 citrus rootstocks in combination with 'Tahiti' acid lime (*Citrus* × *latifolia* (Yu. Tanaka) Tanaka) under irrigation with waters of 0.14 (**A**), 2.40 (**B**), and 4.80 dS m$^{-1}$ (**C**), in the first year. For identification of genotypes, see Table 1. Boxplots with the same lowercase letter do not differ statistically, according to the Tukey test between salinity levels ($p \leq 0.05$), and those with the same uppercase letter belong to the same genotype group, according to the Scott–Knott test ($p \leq 0.05$).

As for the distinction between genotypes used as rootstocks in salinity levels, differentiation is highlighted when irrigated with water of 2.4 dS m$^{-1}$, with the formation of three groups of genotypes, highlighting the 'Santa Cruz Rangpur' lime and TSKFL × CTTR—013 as materials that conferred the highest average weight of fruit to the plants, thus maintaining the quality.

When analyzing the production of the second year (Table 5), there were effects of the interaction between rootstocks and water salinity levels on the number of fruits per plant (NFPL), fruit production (WFPL), and average fruit weight (AFW) ($p \leq 0.05$). When considering the factors independently, significant effects were not observed only for the genotypes used as rootstocks in the average fruit weight (AFW) ($p \leq 0.01$), whereas salinity ($p \leq 0.01$) caused effects on all production variables studied.

**Table 5.** Summary of analysis of variance of number of fruits per plant (NFPL), weight of fruits per plant (WFPL) (g plant$^{-1}$), and average fruit weight (AFW) (g fruit$^{-1}$) of combinations of 'Tahiti' acid lime (*Citrus* × *latifolia* (Yu. Tanaka) Tanaka) with 13 rootstocks under water salinity, at 720 days after the onset of saline water irrigation.

| Variation Factors | DF | Mean Squares | | |
|---|---|---|---|---|
| | | NFPL | WFPL | AFW (g) |
| Block | 3 | 456.65 * | 1,003,605.85 ns | 229.155975 * |
| Genotype (Gen) | 12 | 950.14 ** | 2,213,721.24 ** | 104.240321 ns |
| Error 1 | 36 | 110.48 | 284,424.46 | 68.466528 |
| Salinity (Sal) | 2 | 19,320.75 ** | 54,851,922.32 ** | 861.357813 ** |
| Gen × Sal | 24 | 528.51 ** | 1,524,565.20 ** | 109.237338 ** |
| Error 2 | 78 | 127.10 | 323,892.75 | 39.697871 |
| CV 1 (%) | | 27.60 | 31.20 | 18.81 |
| CV 2 (%) | | 29.61.78 | 33.29 | 14.32 |
| Mean | | 38.077 | 1709.559 | 44.0009814 |

ns = not significant; * and ** = significant at 0.05 and 0.01 probability levels, respectively; CV = coefficient of variation; DF = degrees of freedom; Gen = combination of scion/rootstock.

As occurred in the first year of production, the number of fruits per plant was reduced by the increase in ECw (Figure 4), and the means grouping test showed a higher number of fruits when the rootstocks were 'Sunki Tropical' mandarin and the hybrid TSKC × CTTR—012, under lower salinity (0.14 dS m$^{-1}$) (Figure 4A). It can also be verified that there was no significant difference between the other rootstocks studied, and the hybrid TSKC × CTARG led to a lower average number of fruits per plant, while 'Rangpur Santa Cruz' lime and the citrandarin TSKC × TRBK—007 were related to the lower variability in the production of the second year.

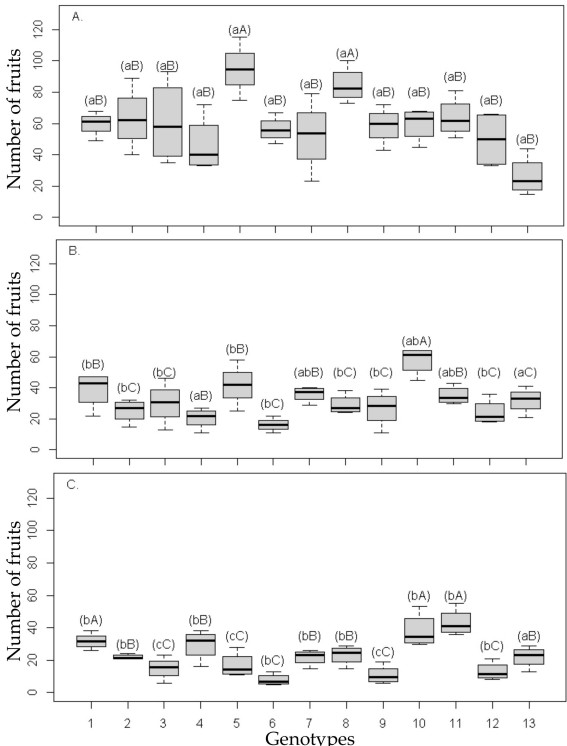

**Figure 4.** Boxplot relative to the average number of fruits per plant of 13 citrus rootstocks in combination with 'Tahiti' acid lime (*Citrus* × *latifolia* (Yu. Tanaka) Tanaka) under irrigation with waters of 0.14 (**A**), 2.40 (**B**), and 4.80 dS m$^{-1}$ (**C**), in the secund year. For identification of genotypes, see Table 1. Boxplots with the same lowercase letter do not differ statistically, according to the Tukey test between salinity levels ($p \leq 0.05$), and those with the same uppercase letter belong to the same genotype group, according to the Scott–Knott test ($p \leq 0.05$).

Application of waters with 2.4 dS m$^{-1}$ (Figure 4B) caused, in addition to the overall reduction in the number of fruits per plant, the formation of three groups of genotypes, with the highest means observed when the rootstock was the citrangor HTR—069. The TSKC × TRBK—007 and TSKFL × TRBK—030 citrandarins, TSKC × CTTR—012 citrangedarin, and TSKC × (LCR × TR)—040 citrimoniandarin led to lower variability in production in the second year of cultivation.

Water with salinity of 4.8 dS m$^{-1}$ caused reductions in the number of fruits in all genotypes (Figure 4C). However, as in the first year of production, lower reductions were observed when the rootstocks were HTR—069 and TSKC × (LCR × TR)—040, which indicates that these rootstocks are better indicated for 'Tahiti' acid lime under salinity.

The weight of fruits per plant (Figure 5) was also reduced by the increase in water salinity in all scion/rootstock combinations, and it was possible to identify three distinct groups of combinations at the three salinity levels.

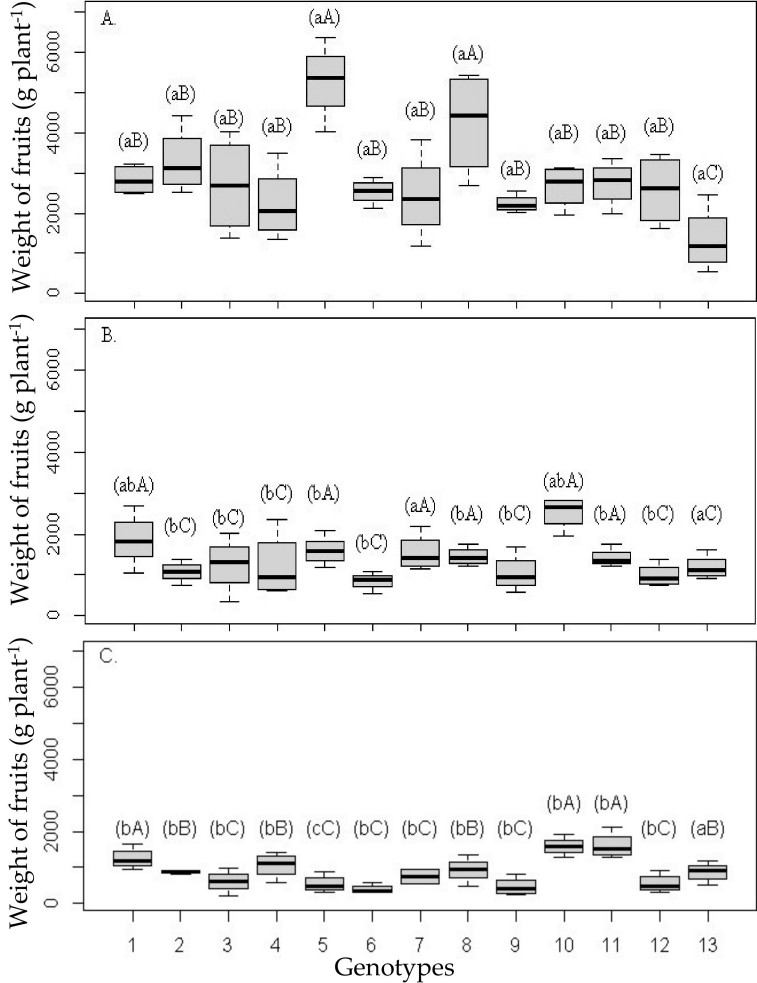

**Figure 5.** Boxplot relative to the average weight of fruits per plant of 13 citrus rootstocks in combination with 'Tahiti' acid lime (*Citrus* × *latifolia* (Yu. Tanaka) Tanaka) under irrigation with waters of 0.14 (**A**), 2.40 (**B**), and 4.80 dS m$^{-1}$ (**C**), in the second year. For identification of genotypes, see Table 1. Boxplots with the same lowercase letter do not differ statistically, according to the Tukey test between salinity levels ($p \leq 0.05$), and those with the same uppercase letter belong to the same genotype group, according to the Scott–Knott test ($p \leq 0.05$).

Regarding irrigation with water of 0.14 dS m$^{-1}$ (Figure 5A), the highest means of the weight of fruits per plant were observed for the rootstocks 'Sunki Tropical' mandarin and TSKC × CTTR—012 citrangedarin.

When the water of 2.4 dS m$^{-1}$ was used in irrigation (Figure 5B), higher means of weight of fruits per plant were observed for 'Santa Cruz Rangpur' lime, 'Sunki Tropical' mandarin, TSKFL × TRBK—030, TSKC × CTTR—012, HTR—069, and TSKC × (LCR × TR)—040.

Conversely, when applying water of 4.8 dS m$^{-1}$ (Figure 5C), the rootstocks 'Rangpur Santa Cruz' lime, HTR—069, and TSKC × (LCR × TR)—040 remained in the group of higher means according to the Scott–Knott test ($p \leq 0.05$), especially the trifoliate hybrid HTR—069, which showed lower variability, proving to have a good level of salinity tolerance.

The mean weight of fruits in the second year of production was significantly reduced by the salinity of the water in some genotypes, different from what occurred in the first year (Figure 6A–C), with the greatest reductions in the weight of the fruits being verified. However, the 'Tahiti' acid lime grafted on 'San Diego' and TSKFL × TRBK—030 citrandarins were also highlighted, which were sensitive to salinity in the first year, too, in addition to plants grafted on 'Sunki Tropical' mandarin, TSKC × CTTR—012, and TSKC × (LCR × TR)—059, with an estimated mean reduction of 50 g fruit$^{-1}$ when irrigated with water of 0.14 dS m$^{-1}$ (Figure 6A) and a mean fruit mass between 35 and 40 g when they were irrigated with water of 4.8 dS m$^{-1}$ (Figure 6C).

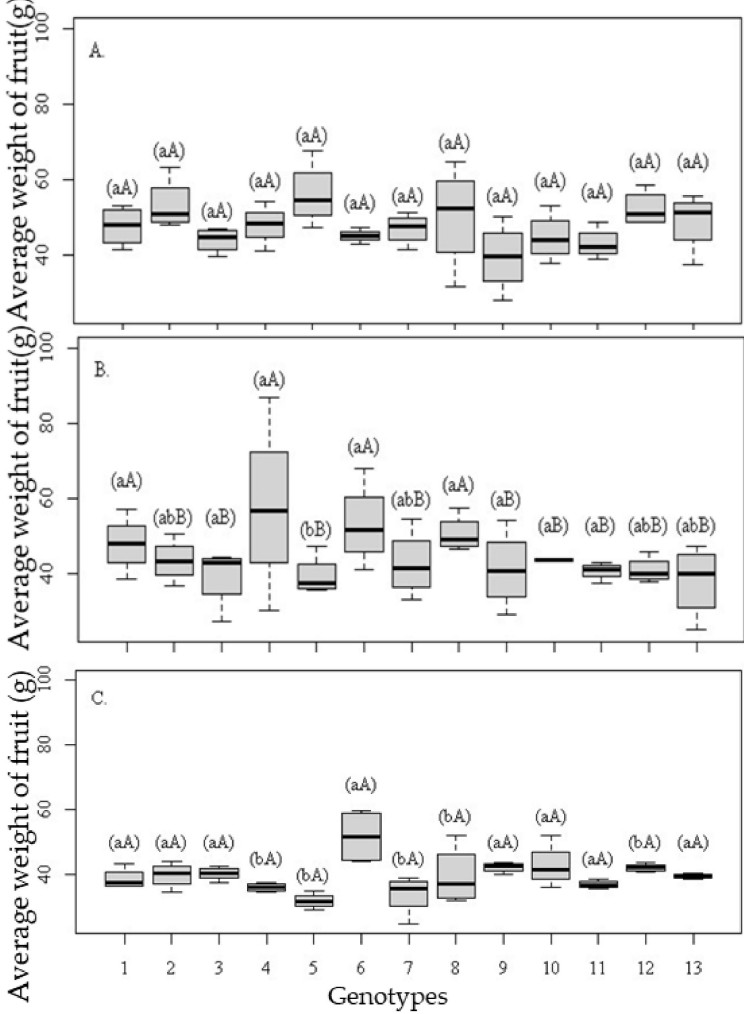

**Figure 6.** Boxplot relative to the average weight of fruit of 13 citrus rootstocks in combination with 'Tahiti' acid lime (*Citrus × latifolia* (Yu. Tanaka) Tanaka) under irrigation with waters of 0.14 (**A**), 2.40 (**B**), and 4.80 dS m$^{-1}$ (**C**), in the second year. For identification of genotypes, see Table 1. Boxplots with the same lowercase letter do not differ statistically, according to the Tukey test between salinity levels ($p \leq 0.05$), and those with the same uppercase letter belong to the same genotype group, according to the Scott–Knott test ($p \leq 0.05$).

The matrices of the analytical performance of the production data with the soil analyses (Figure 7A,B) show the correlations of the first and second years of cultivation, where it is possible to observe negative numbers (without highlighting) related to the negative correlation and positive numbers (highlighted in bold) related to the positive correlation. It was also observed that the higher value means the prediction of the correlation between the variables; for example, the increase in ECse had a positive and predictive correlation with $Ca^{2+}$, $Mg^{2+}$, and $Na^+$ contents in both years, while on the other hand, ECse had a negative and predictive correlation with production variables. However, highlighting the behavior of $K^+$ in the soil during the first and the second years, correlations were predictive with ECse, but positive in the first year and negative in the second year.

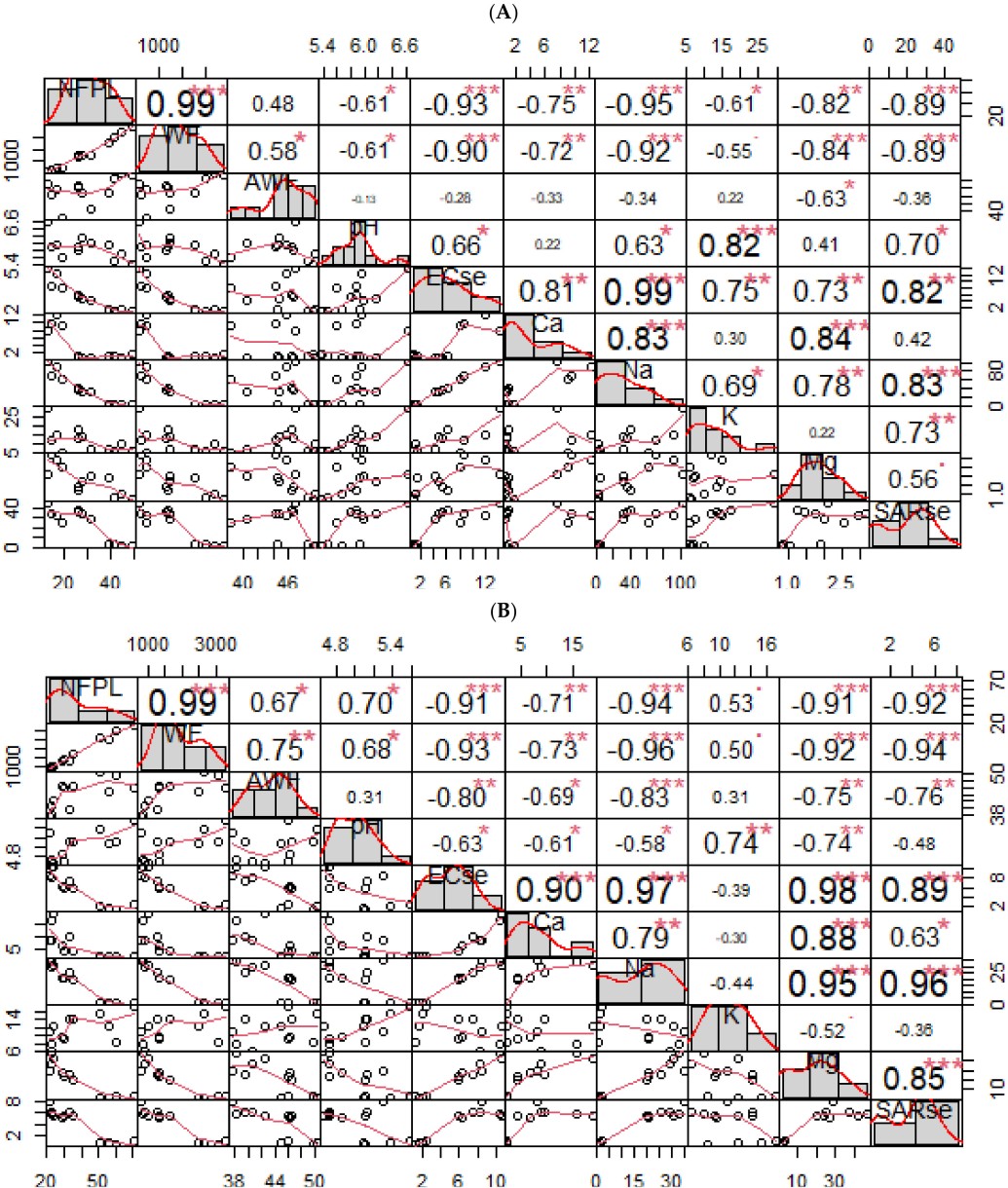

**Figure 7.** Analytical performance of Pearson correlation matrix of production variables (number of fruits per plant (NFPL), weight of fruits (WF), and average weight of fruit (AWF)) of genotypes studied under different water salinity levels, with the chemical characteristics of the soil saturation extract (pH, $Ca^{+2+}$, $Mg^{2+}$, $Na^+$, $K^+$, ECes, and SARse) in the first (**A**) and second years (**B**). ·, *, ** and *** = significant at 0.1, 0.05, 0.01 and 0.001 probability level, respectively.

In the first year, a positive correlation was observed between $K^+$ (0.75) and ECse. However, the opposite was observed in the second year, with a negative correlation ($-0.39$). In the two years of cultivation, the interaction between water salinity and the soil variables pH, ECse, $Ca^{2+}$, $Mg^{2+}$, and SARse was positive and predictive, with a very strong correlation, close to 1, as water with higher electrical conductivity was employed in irrigation.

### 3.3. Chlorophyll a Fluorescence Analysis

The interaction between rootstocks and water salinity levels did not affect chlorophyll *a* fluorescence analysis after adaptation to the dark, and no significant differences were observed among rootstocks or between salinity levels in initial fluorescence ($F_0$), maximum fluorescence (Fm), variable fluorescence (Fv), and quantum efficiency of photosystem II (Fv/Fm), evaluated at 270 days after the beginning of stress (Table 6).

**Table 6.** Summary of analysis of variance of chlorophyll *a* fluorescence analysis on dark stage: initial fluorescence (F0), maximum fluorescence (Fm), variable fluorescence (Fv), and quantum efficiency of photosystem II (Fv/Fm), of combinations of 'Tahiti' acid lime (*Citrus × latifolia* (Yu. Tanaka) Tanaka) with 13 rootstocks under water salinity, at 270 days after the onset of saline water irrigation.

| Variation Factors | DF | Mean Squares | | | |
|---|---|---|---|---|---|
| | | $F_0$ | Fm | Fv | Fv/Fm |
| Block | 3 | 164,284.95 ** | 1,855,164.769 ** | 938,945.117 ** | 0.0048 ns |
| Genotypes (Gen) | 12 | 3035.368 ns | 64,857.381 ns | 54,188.286 ns | 0.0007 ns |
| Error 1 | 36 | 4167.117 | 74,885.949 | 79,660.237 | 0.0018 |
| Salinity (Sal) | 2 | 4458.083 ns | 139,959.480 ns | 172,505.237 ns | 0.0043 ns |
| Gen × Sal | 24 | 3631.208 ns | 49,066.730 ns | 46,836.771 ns | 0.0011 ns |
| Error 2 | 78 | 3942.771 | 62,927.730 | 58,535.619 | 0.0013 |
| CV 1 (%) | | 14.14 | 12.99 | 17.11 | 5.55 |
| CV 2 (%) | | 13.76 | 11.91 | 14.67 | 4.71 |
| Mean | | 456.4679 | 2105.9230 | 1649.4551 | 0.7812 |

ns = not significant; ** = significant at 0.01 probability levels, respectively; CV = coefficient of variation; DF = degrees of freedom; Gen = combination of scion/rootstock.

The fluorescence variables in the second year of cultivation, initial fluorescence ($F_0$), maximum fluorescence (Fm), and variable fluorescence (Fv), showed significant effects caused by the genotype x salinity interaction ($p \leq 0.05$) (Table 7), unlike the first year of cultivation.

**Table 7.** Summary of analysis of variance of chlorophyll *a* fluorescence analysis after dark stage, initial fluorescence (F0), maximum fluorescence (Fm), variable fluorescence (Fv), and quantum efficiency of photosystem II (Fv/Fm), of combinations of 'Tahiti' acid lime (*Citrus × latifolia* (Yu. Tanaka) Tanaka) with 13 rootstocks under water salinity, at 720 days after the onset of saline water irrigation.

| Variation Factors | GL | Mean Squares | | | |
|---|---|---|---|---|---|
| | | $F_0$ | Fm | Fv | Fv/Fm |
| Block | 3 | 1245.21 ns | 328,638.98 ** | 355,262.19 ** | 0.017931 ** |
| Genotype (Gen) | 12 | 3657.48 ns | 67,213.51 ns | 59,530.95 ns | 0.003377 ns |
| Error 1 | 36 | 3116.35 | 52,108.23 | 51,488.36 | 0.004039 |
| Salinity (Sal) | 2 | 2864.95 ns | 33,451.08 ns | 16,915.90 ns | 0.000137 ns |
| Gen × Sal | 24 | 5687.24 * | 111,264.15 * | 86,869.26 * | 0.003540 ns |
| Error 2 | 78 | 3087.59 | 59,136.33 | 51,321.87 | 0.002324 |
| CV 1 (%) | | 14.56 | 14.61 | 19.26 | 8.49 |
| CV 2 (%) | | 14.49 | 15.57 | 19.22 | 6.44 |
| Mean | | 383.53 | 1561.91 | 1178.38 | 0.7481 |

ns = not significant; * and ** = significant at 0.05 and 0.01 probability levels, respectively; CV = coefficient of variation; DF = degrees of freedom; Gen = combination of scion/rootstock.

The effect of salinity on initial fluorescence ($F_0$), maximum fluorescence (Fm), and variable fluorescence (Fv) varied among genotypes. When initial fluorescence was analyzed at the conductivity of 2.4 dS m$^{-1}$ (Figure 8B), two groups of genotypes (scion/rootstock combinations) were formed, with the highest means observed in those in which the rootstocks were 'San Diego' citrandarin, 'Sunki Tropical' mandarin, TSKC × TRBK—030 citrandarin, and TSKFL × CTTR—013 citrangedarin.

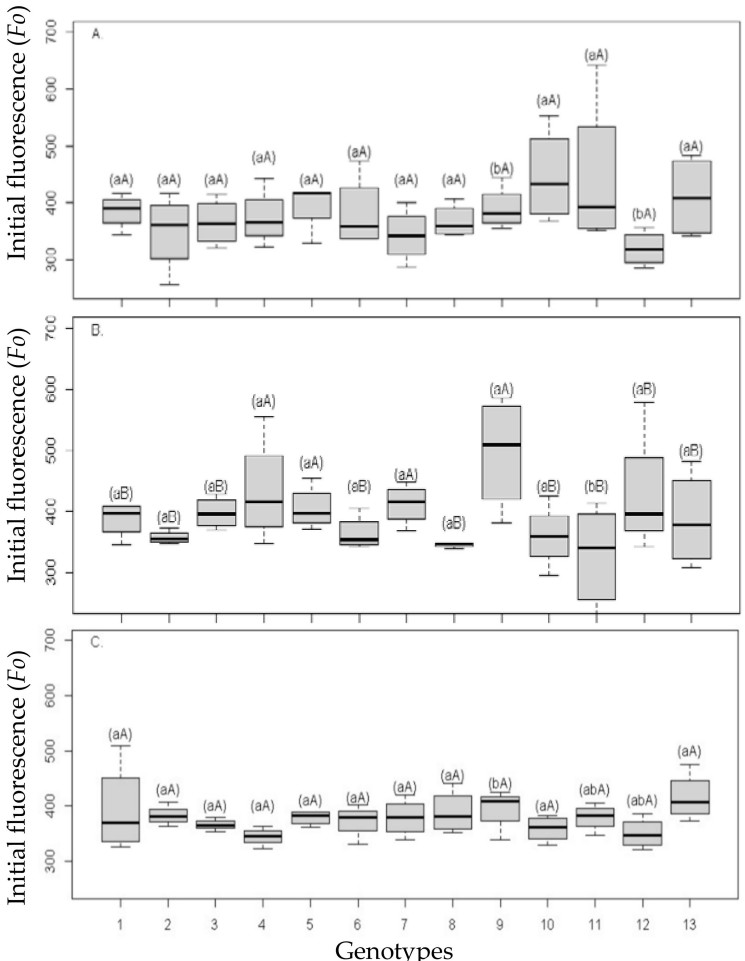

**Figure 8.** Boxplot relative to the average initial fluorescence (F0) of 13 citrus rootstocks in combination with 'Tahiti' acid lime (*Citrus × latifolia* (Yu. Tanaka) Tanaka) under irrigation with waters of 0.14 (**A**), 2.40 (**B**), and 4.80 dS m$^{-1}$ (**C**), in the second year. For identification of genotypes, see Table 1. Boxplots with the same lowercase letter do not differ statistically, according to the Tukey test between salinity levels ($p \leq 0.05$), and those with the same uppercase letter belong to the same genotype group, according to the Scott–Knott test ($p \leq 0.05$).

It was also possible to notice a distinction between salinity levels in plants grafted onto TSKFL × CTTR—013 citrangedarin and TSKC × (LCR × TR)—040 and TSKC × (LCR × TR)—059 citrimoniandarins. The highest values were observed when applying water of 2.4 dS m$^{-1}$ in the combinations of the hybrids TSKFL × CTTR—013 and TSKC × (LCR × TR)—059, and when applying water of 0.14 dS m$^{-1}$ in plants which had the hybrid TSKC × (LCR × TR)—040 as a rootstock (Figure 8A).

The maximum fluorescence of plants (Fm) (Figure 9) was significantly reduced only in 'Tahiti' plants grafted onto TSKC × (LCR × TR)—040 citrimoniandarin and TSKC × CTARG—019 citrangedarin, especially when they were irrigated with waters of 4.8 dS m$^{-1}$, highlighting this variable as an indicator of ionic stress.

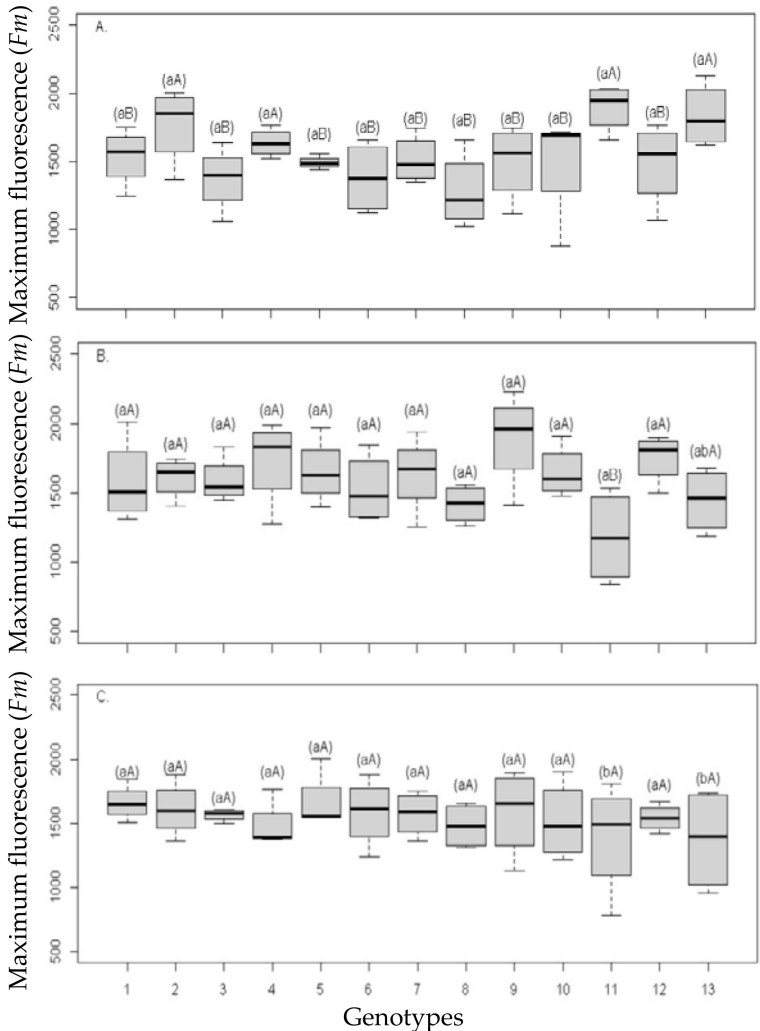

**Figure 9.** Boxplot relative to the average maximum fluorescence (Fm) of 13 citrus rootstocks in combination with 'Tahiti' acid lime (*Citrus × latifolia* (Yu. Tanaka) Tanaka) under irrigation with waters of 0.14 (**A**), 2.40 (**B**), and 4.80 dS m$^{-1}$ (**C**), in the second year. For identification of genotypes, see Table 1. Boxplots with the same lowercase letter do not differ statistically, according to the Tukey test between salinity levels ($p \leq 0.05$), and those with the same uppercase letter belong to the same genotype group, according to the Scott–Knott test ($p \leq 0.05$).

Variable fluorescence (Fv) was different among the genotypes only at the lowest level of water salinity, highlighting two groups of genotypes (Figure 10). However, the increase in water salinity reduced ($p \leq 0.05$) the variable fluorescence in 'Tahiti' plants grafted onto the hybrids TSKC × (LCR× TR)—040 and TSKC × CTARG—019, which could be related to the reduction in the maximum fluorescence values observed in these genotypes and to the increase in the minimum fluorescence values recorded in TSKC × (LCR × TR)—040 citrimoniandarin.

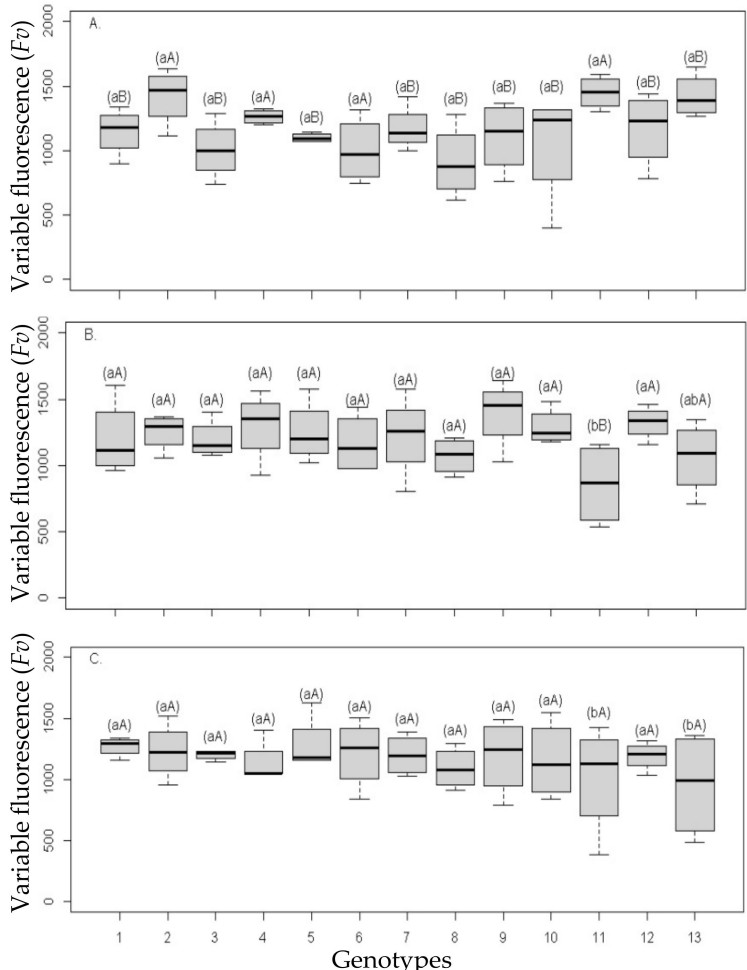

**Figure 10.** Boxplot relative to the average variable fluorescence (Fv) of 13 citrus rootstocks in combination with 'Tahiti' acid lime (*Citrus* × *latifolia* (Yu. Tanaka) Tanaka) under irrigation with waters of 0.14 (**A**), 2.40 (**B**), and 4.80 dS m$^{-1}$ (**C**), in the second year. For identification of genotypes, see Table 1. Boxplots with the same lowercase letter do not differ statistically, according to the Tukey test between salinity levels ($p \leq 0.05$), and those with the same uppercase letter belong to the same genotype group, according to the Scott–Knott test ($p \leq 0.05$).

### 3.4. Gas Exchange

There was a significant effect ($p \leq 0.05$) of the interaction between rootstocks and water salinity levels (Table 8) on the $CO_2$ assimilation rate (*A*), stomatal conductance (*gs*), transpiration (*E*), and intrinsic carboxylation efficiency (*CEi*) (Table 8). Similarly, there was also a significant effect ($p \leq 0.01$) of the salinity factor on the internal $CO_2$ concentration (*Ci*), with no significant effect of any factor on the intrinsic water use efficiency (*WUEi*).

For the genotype factor, significant effects were found only in the variables *A*, *E*, *gs*, and *CEi* ($p \leq 0.01$), highlighting that water salinity affects citrus rootstocks differently in most of the gas exchange variables studied.

The $CO_2$ assimilation rate (*A*) varied among the 13 scion/rootstock combinations according to the water salinity level (Figure 11), leading to the formation of 3 groups of genotypes when the plants were irrigated with waters of 0.14 dS m$^{-1}$ (Figure 11A) and 2 groups of genotypes when the plants were irrigated with waters of 2.40 dS m$^{-1}$ (Figure 11B) and 4.80 dS m$^{-1}$ (Figure 11C) according to the Scott–Knott test ($p \leq 0.05$). However, it was found that salinity only reduced the net photosynthesis of 'Tahiti' plants grafted onto 'Santa Cruz Rangpur' lime, 'Riverside' citrandarin, TSKC × CTTR—012 and TSKC × CTARG—019 citrangedarins, and TSKC × (LCR × TR)—040 citrimoniandarin.

**Table 8.** Summary of analysis of variance of gas exchange—assimilation rate ($A$) ($\mu mol\ m^{-2}\ s^{-1}$), transpiration ($E$) ($mmol_{H2O}\ m^{-2}\ s^{-1}$), stomatal conductance ($gs$) ($mol_{H2O}\ m^{-2}\ s^{-1}$), $CO_2$ concentration ($Ci$) ($\mu mol\ mol^{-1}$), intrinsic water use efficiency ($WUEi$) (($\mu mol_{CO2}\ m^{-2}\ s^{-1}$) ($mmol_{H2O}\ m^{-2}\ s^{-1})^{-1}$), and intrinsic carboxylation efficiency ($CEi$) ($\mu mol_{CO2}\ m^{-2}\ s^{-1}$) of combinations of 'Tahiti' acid lime (*Citrus × latifolia* (Yu. Tanaka) Tanaka) with 13 rootstocks under water salinity at 270 days after the onset of saline water irrigation.

| Variation Factors | DF | Mean Squares | | | | | |
|---|---|---|---|---|---|---|---|
| | | $A$ | $E$ | $gs$ | $Ci$ | $WUEi$ | $CEi$ |
| Block | 3 | 15.5781 ** | 0.1015 ns | 0.0057 ** | 6624.621 ** | 11.456 ns | 0.00006 ns |
| Genotype (Gen) | 12 | 13.4054 ** | 0.4518 ** | 0.0039 ** | 1116.853 ns | 6.1785 ns | 0.0002 ** |
| Error 1 | 36 | 1.6734 | 0.1565 | 0.0006 | 1102.876 | 6.4194 | 0.00004 |
| Salinity (Sal) | 2 | 13.0319 ** | 0.5770 ** | 0.0030 ** | 2166.160 * | 1.3231 ns | 0.0002 ** |
| Gen × Sal | 24 | 3.4415 ** | 0.1569 * | 0.0006 ** | 398.861 ns | 4.9946 ns | 0.00007 ** |
| Error 2 | 78 | 1.3732 | 0.0884 | 0.0003 | 510.670 | 6.0904 | 0.00003 |
| CV 1 (%) | | 21.86 | 30.53 | 38.42 | 15.32 | 51.97 | 24.99 |
| CV 2 (%) | | 19.80 | 22.96 | 27.09 | 10.42 | 50.62 | 21.98 |
| Mean | | 5.9188 | 1.2955 | 0.0641 | 216.8397 | 4.8753 | 0.0275 |

ns = not significant; * and ** = significant at 0.05 and 0.01 probability levels, respectively; CV = coefficient of variation; DF = degrees of freedom; Gen = combination of scion/rootstock.

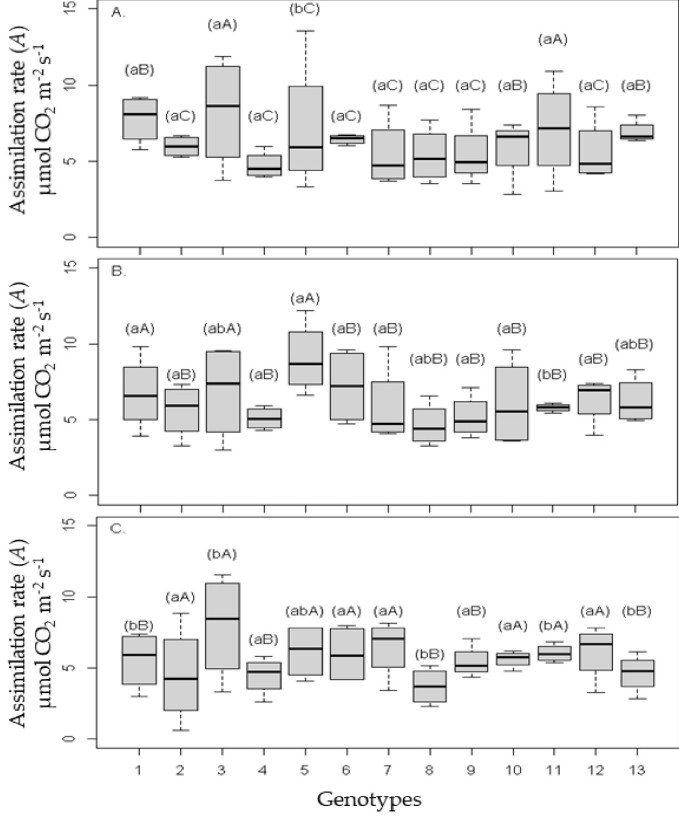

**Figure 11.** Boxplot relative to the assimilation rate ($A$) of 13 citrus rootstocks in combination with 'Tahiti' acid lime (*Citrus × latifolia* (Yu. Tanaka) Tanaka) under irrigation with waters of 0.14 (**A**), 2.40 (**B**), and 4.80 dS m$^{-1}$ (**C**), in the first year. For identification of genotypes, see Table 1. Boxplots with the same lowercase letter do not differ statistically, according to the Tukey test between salinity levels ($p \leq 0.05$), and those with the same uppercase letter belong to the same genotype group, according to the Scott–Knott test ($p \leq 0.05$).

When irrigation was performed with waters of 4.8 dS m$^{-1}$, it was found that 'Indio' citrandarin, 'Sunki Tropical' mandarin, and the hybrids TSKC × TRBK—007, TSKFL × TRBK—030,

HTR—069, and TSKC × (LCR × TR)—059 led to a higher value of assimilation rate in the scion variety, being grouped among the genotypes with the highest means.

On the other hand, the largest reductions in assimilation rate ($A$) were noted for Tahiti grafted on 'Rangpur Santa Cruz' lime, 'Riverside' citrandarin, TSKC × CTTR—012 citrangedarin, TSKC × (CSF × TR)—040 citrimoniandarin, and TSKC × CTARG—019 citrangedarin.

Figure 12 contains the boxplots related to the means of transpiration ($E$) of the 13 scion/rootstock combinations under irrigation with waters of 0.14 dS m$^{-1}$ (Figure 12A), 2.40 dS m$^{-1}$ (Figure 12B), and 4.80 dS m$^{-1}$ (Figure 12C), and it is possible to verify the rootstocks 'Santa Cruz Rangpur' lime and the 'Riverside' citrandarin between the genotypes with the highest mean values at all salinity levels. However, the increase in salinity, in general, caused a reduction in plant transpiration, but this reduction was more significant when water with an ECw of 4.8 dS m$^{-1}$ was applied to the hybrids TSKC × CTTR—012, TSKC × (LCR × TR)—040, and TSKC × CTARG—019, and the rootstock 'Santa Cruz Rangpur' lime.

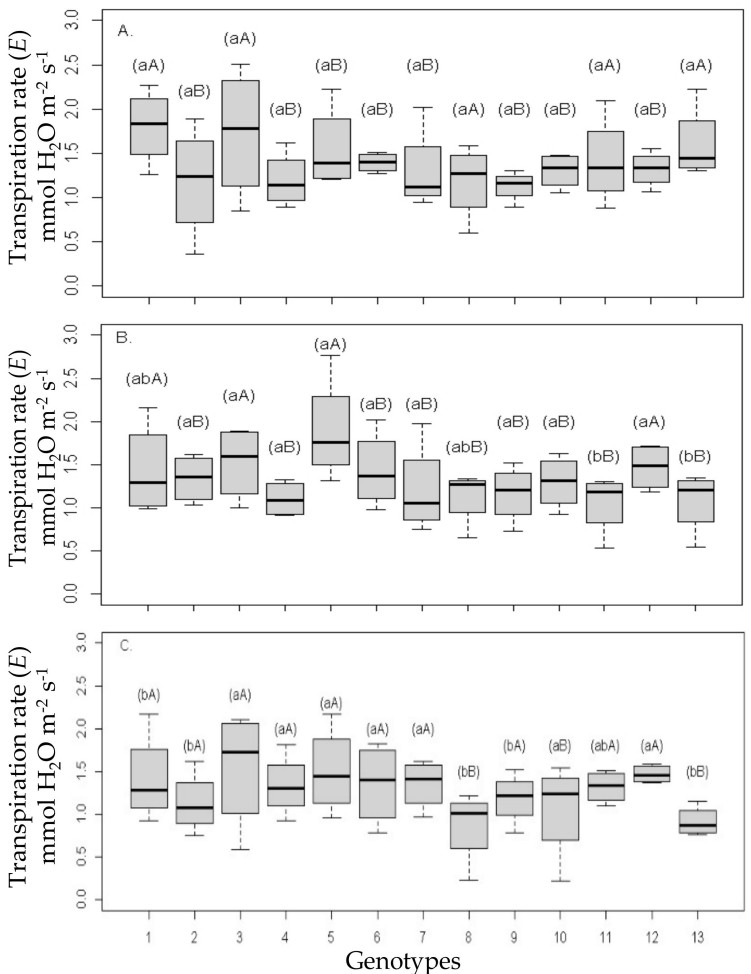

**Figure 12.** Boxplot relative to the average transpiration rate ($E$) of 13 citrus rootstocks in combination with 'Tahiti' acid lime (*Citrus* × *latifolia* (Yu. Tanaka) Tanaka) under irrigation with waters of 0.14 (**A**), 2.40 (**B**), and 4.80 dS m$^{-1}$ (**C**), in the first year. For identification of genotypes, see Table 1. Boxplots with the same lowercase letter do not differ statistically, according to the Tukey test between salinity levels ($p \leq 0.05$), and those with the same uppercase letter belong to the same genotype group, according to the Scott–Knott test ($p \leq 0.05$).

Stomatal conductance (*gs*) (Figure 13) was reduced by salinity differently among the rootstocks, and higher values of *gs* mean that the stomata of the plant are more open, which allows the influx of $CO_2$ and, consequently, substrate for photosynthesis. In this context, although 'Santa Cruz Rangpur' lime and 'Riverside' citrandarin were in the group of the highest means when irrigated with waters of 0.14 dS m$^{-1}$ (Figure 13A), when irrigated with water of 4.8 dS m$^{-1}$ (Figure 13C), they showed reductions of the order of 54.5% and 30.8% in *gs* values, respectively.

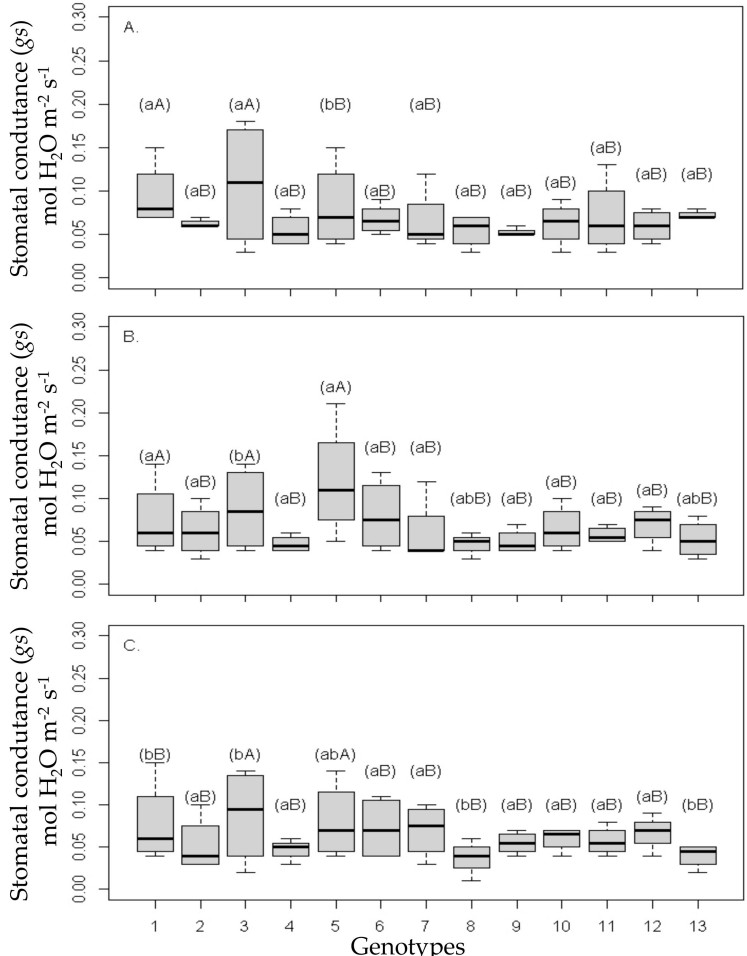

**Figure 13.** Boxplot relative to the average stomatal conductance (*gs*) of 13 citrus rootstocks in combination with 'Tahiti' acid lime (*Citrus × latifolia* (Yu. Tanaka) Tanaka) under irrigation with waters of 0.14 (**A**), 2.40 (**B**), and 4.80 dS m$^{-1}$ (**C**), in the first year. For identification of genotypes, see Table 1. Boxplots with the same lowercase letter do not differ statistically, according to the Tukey test between salinity levels ($p \leq 0.05$), and those with the same uppercase letter belong to the same genotype group, according to the Scott–Knott test ($p \leq 0.05$).

The evaluation of gas exchange (Table 9) in the second year of cultivation revealed a significant effect of the interaction between rootstocks and salinity levels only for stomatal conductance (*gs*) ($p \leq 0.01$). For the genotype factor, there was no significant effect in any of the gas exchange variables. Salinity as an isolated factor affected the $CO_2$ assimilation rate (*A*) and transpiration (*E*), indicating, besides the ionic effect already observed in fluorescence reactions, osmotic effects on the plants, a situation that confirms the stress condition.

**Table 9.** Summary of analysis of variance of gas exchange—assimilation rate ($A$) ($\mu$mol m$^{-2}$ s$^{-1}$), transpiration ($E$) (mmol$_{H2O}$ m$^{-2}$ s$^{-1}$), stomatal conductance ($gs$) (mol$_{H2O}$ m$^{-2}$ s$^{-1}$), $CO_2$ concentration ($Ci$) ($\mu$mol mol$^{-1}$), intrinsic water use efficiency ($WUEi$) (($\mu$mol$_{CO2}$ m$^{-2}$ s$^{-1}$) (mmol$_{H2O}$ m$^{-2}$ s$^{-1}$)$^{-1}$), and intrinsic carboxylation efficiency ($CEi$) ($\mu$mol$_{CO2}$ m$^{-2}$ s$^{-1}$) of combinations of 'Tahiti' acid lime (*Citrus* $\times$ *latifolia* (Yu. Tanaka) Tanaka) with 13 rootstocks under water salinity at 720 days after the onset of saline water irrigation.

| Variation Factors | DF | Mean Squares | | | | | |
|---|---|---|---|---|---|---|---|
| | | $A$ | $E$ | $gs$ | $Ci$ | $WUEi$ | $CEi$ |
| Block | 3 | 0.0935 $^{ns}$ | 0.0009 $^{ns}$ | 1.6405 ** | 14.9985 * | 1.0832 ** | 0.0015 $^{ns}$ |
| Genotype (Gen) | 12 | 0.0632 $^{ns}$ | 0.0006 $^{ns}$ | 0.3186 $^{ns}$ | 7.8753 $^{ns}$ | 0.0833 $^{ns}$ | 0.0012 $^{ns}$ |
| Error 1 | 36 | 0.0423 | 0.0037 | 0.3597 | 4.7952 | 0.1037 | 0.0014 |
| Salinity (Sal) | 2 | 0.1704 ** | 0.0011 * | 0.8505 $^{ns}$ | 0.0513 $^{ns}$ | 0.0813 $^{ns}$ | 0.0003 $^{ns}$ |
| Gen $\times$ Sal | 24 | 0.0494 $^{ns}$ | 0.0004 $^{ns}$ | 0.6301 ** | 4.3351 $^{ns}$ | 0.1114 $^{ns}$ | 0.0012 $^{ns}$ |
| Error 2 | 78 | 0.0329 | 0.0003 | 0.2848 | 3.5694 | 0.0988 | 0.0009 |
| CV 1 (%) | | 12.07 | 1.86 | 19.66 | 17.26 | 13.90 | 3.62 |
| CV 2 (%) | | 10.65 | 1.61 | 17.50 | 14.89 | 13.56 | 2.87 |
| Mean | | 1.7043 | 1.0364 | 3.0500 | 12.6887 | 2.3168 | 1.0327 |

$^{ns}$ = not significant; * and ** = significant at 0.05 and 0.01 probability levels, respectively; CV = coefficient of variation; DF = degrees of freedom; Gen = combination of scion/rootstock.

The $CO_2$ assimilation rates ($A$) were not statistically different among the 13 scion/rootstock combinations at any of the salinity levels of irrigation water (Figure 14A–C), i.e., no groups of genotypes with higher means were formed. However, the effect of salinity was different among the genotypes, with a reduction in net photosynthesis in 'Tahiti' plants grafted onto the hybrid TSKC $\times$ CTARG—019, indicative of its sensitivity since decreases were observed in gas exchange and fluorescence variables.

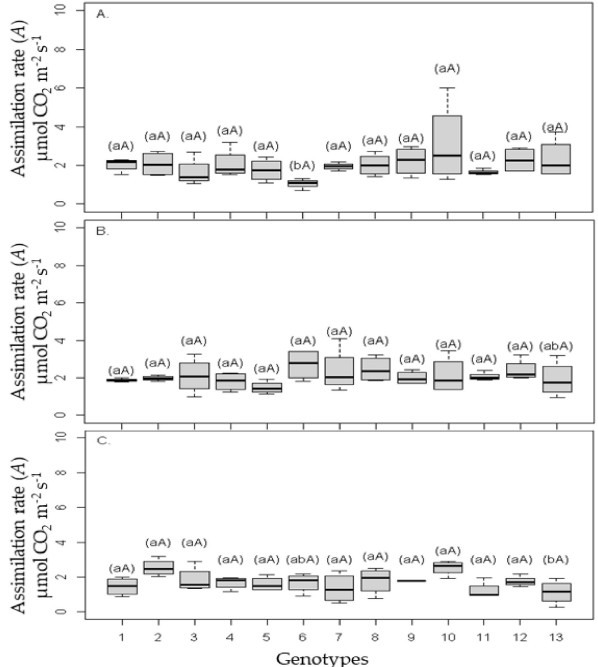

**Figure 14.** Boxplot relative to the assimilation rate ($A$) of 13 citrus rootstocks in combination with 'Tahiti' acid lime (*Citrus* $\times$ *latifolia* (Yu. Tanaka) Tanaka) under irrigation with waters of 0.14 (**A**), 2.40 (**B**), and 4.80 dS m$^{-1}$ (**C**), in the second year. For identification of genotypes, see Table 1. Boxplots with the same lowercase letter do not differ statistically, according to the Tukey test between salinity levels ($p \leq 0.05$), and those with the same uppercase letter belong to the same genotype group, according to the Scott–Knott test ($p \leq 0.05$).

There was no difference ($p > 0.05$) between rootstocks in terms of transpiration rate (*E*) and $CO_2$ assimilation rate of 'Tahiti' acid lime (Figure 15A–C). Even with the application of saline water with electrical conductivity levels of 2.4 and 4.8 dS m$^{-1}$, the recorded values of *E* were between 1.0 and 3.0 mmol $H_2O$ m$^{-2}$ s$^{-1}$, except for the values observed in 'Indio' citrandarin, which showed greater variation under an ECw of 4.8 dS m$^{-1}$. The reduction in transpiration values should be highlighted, especially in plants grafted onto TSKC × CTARG—019 citrangedarin.

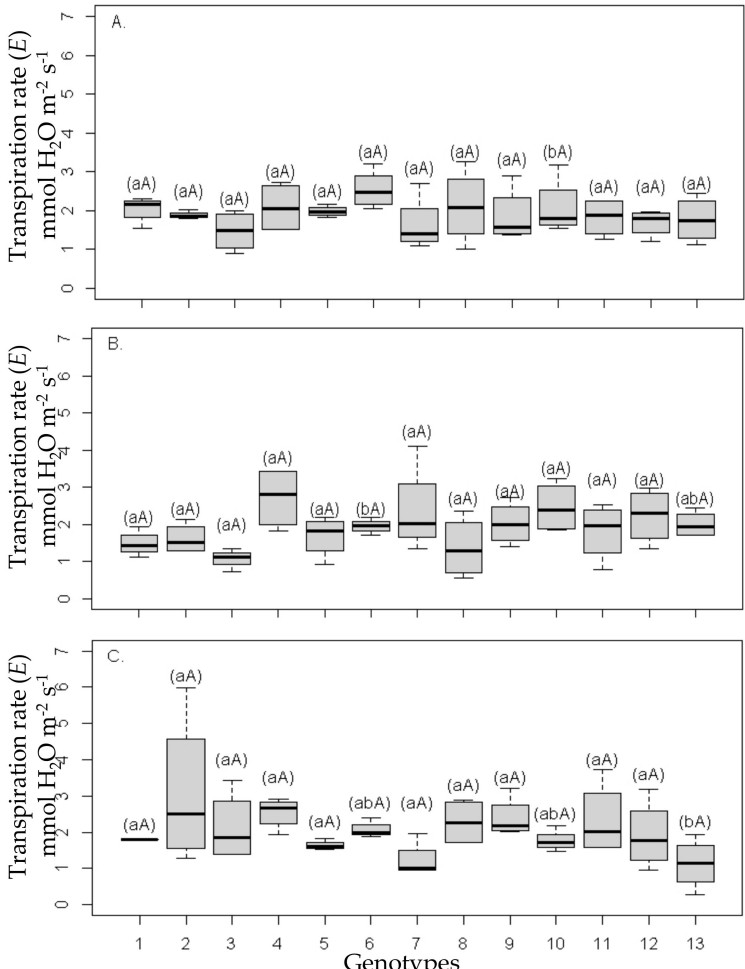

**Figure 15.** Boxplot relative to the average transpiration rate (*E*) of 13 citrus rootstocks in combination with 'Tahiti' acid lime (*Citrus* × *latifolia* (Yu. Tanaka) Tanaka) under irrigation with waters of 0.14 (**A**), 2.40 (**B**), and 4.80 dS m$^{-1}$ (**C**), in the second year. For identification of genotypes, see Table 1. Boxplots with the same lowercase letter do not differ statistically, according to the Tukey test between salinity levels ($p \leq 0.05$), and those with the same uppercase letter belong to the same genotype group, according to the Scott–Knott test ($p \leq 0.05$).

Stomatal conductance (*gs*) did not vary between the rootstocks at each water salinity level applied (Figure 16), but it was observed that the effect of salinity was different among genotypes, especially because there was a significant reduction in the values of stomatal conductance in the hybrid TSKC × CTARG—019 under an ECw of 4.8 dS m$^{-1}$.

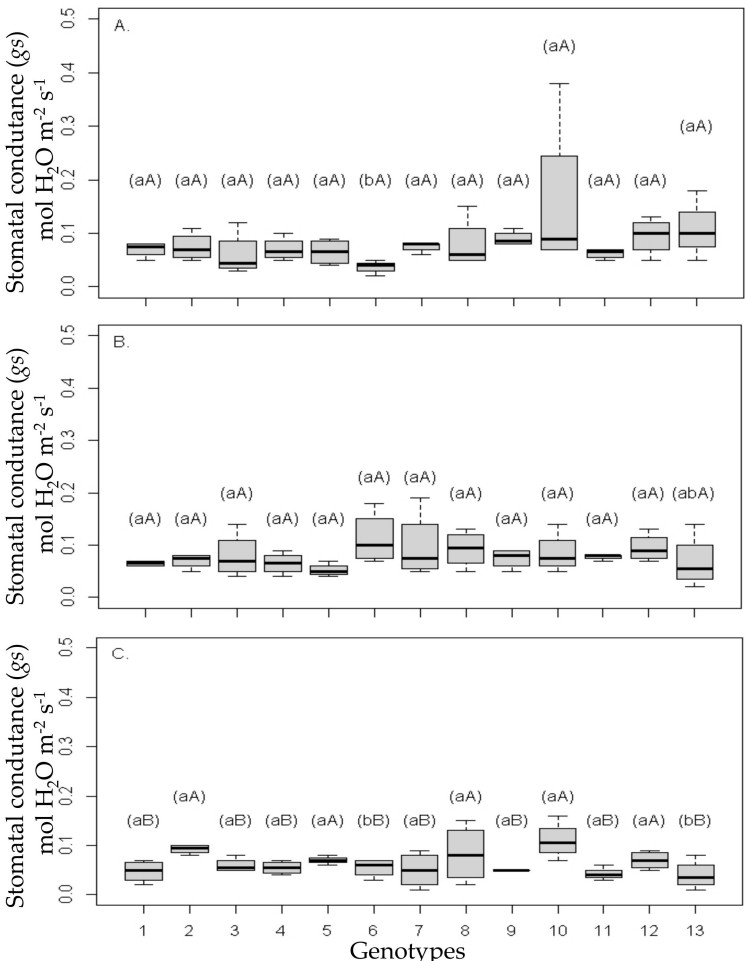

**Figure 16.** Boxplot relative to the average stomatal conductance (*gs*) of 13 citrus rootstocks in combination with 'Tahiti' acid lime (*Citrus × latifolia* (Yu. Tanaka) Tanaka) under irrigation with waters of 0.14 (**A**), 2.40 (**B**), and 4.80 dS m$^{-1}$ (**C**), in the second year. For identification of genotypes, see Table 1. Boxplots with the same lowercase letter do not differ statistically, according to the Tukey test between salinity levels ($p \leq 0.05$), and those with the same uppercase letter belong to the same genotype group, according to the Scott–Knott test ($p \leq 0.05$).

## 4. Discussion

### 4.1. Soil Analysis

Soil analysis in the first year showed an increase in Na$^+$ contents, which can alter soil physical characteristics due to clay dispersion and reduce porosity, hydraulic conductivity, and infiltration rate, as well as causing soil destructuring, thereby affecting plant development [19,20].

However, the reduction in the infiltration rate was not observed in pots used as lysimeters, despite the records of higher sodium contents to the detriment of potassium contents (Table 3).

It should be emphasized that the potassium contents observed in the soil, in all treatments, can be classified as very high [15]. However, the increase in sodium content can cause a nutritional imbalance, since these ions have similar ionic radii, competing for similar exchange sites, making those that are in a greater quantity more available. On the other hand, for citrus plants, only in some genotypes, especially the most sensitive ones, was it possible to observe symptoms characteristic of potassium deficiency and sodium accumulation in the cell vacuole.

The exchangeable sodium percentage and salt content in the soil recorded during the second year allowed classifying the soil as saline, but non-sodic, which can be related to a more significant rainy season, with precipitation in the region slightly above average (751 mm), which even reduced the frequency and duration of saline water irrigation events in addition to the increase in nutrient requirements in the second year of citrus plants (Table 3).

The analysis of soil chemical characteristics in the second year was affected by the reduction in $K^+$ contents, which indicates a greater absorption of this nutrient by plants. In the experiment, the plants well-supplemented with this chemical element showed no symptoms of mineral deficiency.

In the same line, potassium ($K^+$) is an essential element for cell osmotic adjustment, as well as participating in stomatal opening and closure, which influences the water status of the plants and confers greater tolerance to salt stress [21]. Additionally, it is important to observe the decreased pH value of about 1 unit in the second year, characteristics that influence the availability of micronutrients.

*4.2. Production Analysis*

The increase in salt concentration in the soil causes negative effects on the plant, especially the reduction in yield, attributed to several physiological and biochemical processes [22], also observed in the present study in the first year of cultivation (Table 5), but with different behaviors of the rootstocks under salinity. These findings may be related to the distinct responses shown by citrus under salt stress, as salt tolerance varies with species, scion/rootstock combination, and plant development stage [8], under the influence of specific genes and environmental factors [23,24].

The hypothesis presented was verified when water salinity was increased to 4.8 dS m$^{-1}$, which caused reductions in the number of fruits, during the first year, in all genotypes (Figure 1C). However, in genotypes 5, 6, 8, 10, and 11, corresponding to 'Sunki Tropical' mandarin, TSKC × TRBK—007 citrandarin, TSKC × CTTR—012 citrangedarin, HTR—069 citrangor, and TSKC × (LCR × TR)—040 citrimoniandarin, lower reductions were observed, which characterized the lower sensitivity of 'Tahiti' acid lime when grafted onto these rootstocks. It is worth highlighting that TSKC × TRBK—007 citrandarin showed greater stability in production when irrigated with water of 4.80 dS m$^{-1}$.

When studying the formation of citrus rootstocks (before grafting) under water salinity, some authors [8,25] state that water of up to 2.0 dS m$^{-1}$ can be used, similar to this study, in which water of electrical conductivity 2.4 dS m$^{-1}$ was used.

In addition, during the first year of production, the number of fruits per plant was not different among the genotypes when irrigation was performed with waters of 2.4 and 4.8 dS m$^{-1}$. However, in 'Tahiti' acid lime plants grafted onto 'Riverside' citrandarin, 'Sunki Tropical' mandarin, and the trifoliate hybrid HTR—069 citrangor, there was greater stability of production between the salinity levels (Figure 2). The results obtained highlight these materials as promising since in the agricultural production system it is of great importance that production is satisfactory in quantity and for a longer time. However, it is also emphasized that such distinctions in the number of fruits per plant may be related to the early production of some genotypes, which shows the importance of having more years of evaluation.

The evaluation of the second production year made it possible to confirm the information obtained in the first year when the distinction of genotypes irrigated with waters of higher salinity levels was verified. The observations can be confirmed particularly for HTR—069, a hybrid of 'Pera' sweet orange with 'Yuma' citrange, as a rootstock that confers greater tolerance to salinity, corroborating the results obtained with this hybrid in the seedling formation stage [10]. On the other hand, highlighting this rootstock does not disqualify the potential of the others, since citrus plants show stabilization of production between six and seven years [15]. Considering that this study looked at the second year of

production and third year of cultivation, the scion/rootstock combinations under analysis should be studied for a longer period.

Regarding the production and number of fruits per plant, the effects of salinity were different among the rootstocks, which points to the need to highlight the variation in the juvenility period of citrus, and it can be verified that the different production may be related to the precocious or later onset of fruiting of one or another scion/rootstock combination. Therefore, it is not possible to accurately predict fruit yield in the future, a situation that requires the monitoring of five or more seasons.

However, when evaluating the effect of salinity levels, it was possible to observe differences between the scion/rootstock combinations regarding the tolerance to this abiotic stress, since less variation in production between salinity levels, for a certain combination, means the obtaining of viable yields, even under stress conditions, which was found when the rootstocks were the hybrids HTR—069 and TSKC × (LCR × TR)—040. This result indicates tolerance to the salinity of these varieties and their potential for cultivation under irrigation in the Brazilian semi-arid region.

The correlations between production variables and salinity levels observed in the soil during the first and second years of production generated information that explains the increase in the need for some nutrients, especially potassium, by plants so that they could maintain physiological and biochemical processes and survival, since the ability of citrus plants to develop in saline soils is generally more associated with exclusion than with compartmentalization of ions in their leaves [26]; that is, more efficient plants avoid the accumulation of toxic ions, reducing the absorption of ions.

Another inference that can be made is that, both in the first and the second year of cultivation, salinity negatively affected the production variables NFPL, WFPL, and AFW (Figure 7A,B); that is, as salinity increased, production decreased, corroborating what was shown in the boxplots presented in Figures 1–6, as well as in studies conducted by several authors [8,10,25,27].

The process of formation of saline soils involves the increase in the concentration of soluble salts in the soil solution and results in the accumulation of $Na^+$, $Ca^{2+}$, $Mg^{2+}$, and $K^+$, so the constant use of saline water must be well-managed to avoid or delay salinization phenomena. In this context, it was verified that the water used in irrigation had high contents of cations, especially $Ca^{2+}$, $Mg^{2+}$, and $Na^+$, which caused an increase in the contents of these elements in the soil and in the relationships between them. The increase in sodium, compared to the others, should be highlighted, since this is the cation in the greatest proportion, as observed in the two years through correlation analyses.

In general, the main salts found were the chlorides and sulfates of $Na^+$, $Ca^{2+}$, and $Mg^{2+}$, however carbonates and nitrates were also found [28,29]. In this work, the water used for irrigation had some elements mentioned, which were reflected in the contents found in the soil cultivated with citrus in the two years, although in a more significant way in the second year.

### 4.3. Chlorophyll a Fluorescence

The effects of salinity on plants may be osmotic or ionic, with the former corresponding to the decrease in osmotic potential and the latter related to nutritional imbalance, due to the high concentration of specific ions, which results in a series of harmful effects on plant physiology, including chlorophyll fluorescence and gas exchange, which are severely affected [18,27].

The effects of salinity on fruit production generated physiological disorders in the first year of cultivation; however, the effects were osmotic since, in the fluorescence analyses performed in the first year, the damage to the photosynthetic apparatus was not significant, indicating that the ionic effect was not deleterious in the scion/rootstock combinations to the point of causing damage to the photosynthetic apparatus, as observed by Zang et al. [30].

In stressful environments, a decrease in the potential quantum efficiency of photosystem II can occur and is detected by the reduction in the Fv/Fm ratio [31]. Although

the plants were irrigated with waters of 4.8 dS m$^{-1}$, which caused an accumulation of salts in the soil of the order of 14.7 dS m$^{-1}$ in the first year and 9.5 dS m$^{-1}$ in the second year, values well above the salinity threshold of the citrus crop (1.4 dS m$^{-1}$), there was no significant effect on the quantum efficiency of photosystem II to the point of causing deleterious effects.

In the process of photosynthesis, the absorbed light can be transferred to the photosystems or, if there is excess energy, it is dissipated in the form of heat or fluorescence [31,32]. Considering that there were no changes in fluorescence with the increase in salt concentration in the irrigation water applied to plants and that values of quantum efficiency of photosystem II (Fv/Fm) were close to 0.78 in the first year (Table 7), it is possible to assume that the photosynthetic apparatus of the scion/rootstock combinations was not damaged in the present study, with possible acclimatization, or that this response was a consequence of the short period of stress, or due to the more regular distribution of rainfall, which caused a decrease in salt concentration since irrigation was performed based on water balance and a leaching fraction of 10% was used, which enabled a reduction in the accumulation of salts in the soil in the second year.

However, according to Barbosa et al. [25] and Fernandes et al. [33], in studies on the growth and physiology of hybrids and some varieties recommended as rootstock for citrus, when plants are irrigated with saline waters, the response may be diverse due to the genetic load of the material, as well as to the ability to adapt to stress conditions, which may have occurred in the present study and thus preserved the PSII. Therefore, it is necessary to conduct studies for longer periods under controlled environments, where ambient conditions can be maintained so as not to cause other stresses or other types of interference.

The photochemical quantum efficiency of PSII (FV/Fm) reflects the efficiency related to the absorption of light energy by the PSII antenna complex and its conversion into chemical energy. Therefore, as there was no significant effect of salinity on this variable, especially because the mean value was higher than 0.72, a limit that indicates damage to the photosynthetic apparatus [19], there was no damage caused by salt stress [34,35].

The intensification of stress, in the second year of plants under irrigation with saline waters, made it possible to observe more significant physiological disorders, as reported by Zang et al. [30], who concluded that the deleterious effects of irrigation water salinity on scion/rootstock combinations may be more intense with longer exposure to stress. This fact was verified in the present study since the salinity in the saturation extract was maintained in the second year (Table 3), and it was possible to notice exhaustion of plants that were under prolonged stress, although there was a period of rainfall.

The most noticeable effect mentioned here is related to the different behaviors of the genotypes regarding chlorophyll fluorescence when irrigated with saline waters, although there were no effects of the factors on the quantum efficiency of photosystem II (Fv/Fm) (Table 8), which can be related to acclimatization, although the plants were under stress.

The hypothesis is that salinity triggered reactions, which were noticed through the increase in the values of initial, maximum, and variable fluorescence, although this does not mean higher or lower sensitivity to salinity, as there may be physiological adjustments that allow the plant to maintain energy formation for metabolic processes, even with lower efficiency.

For the sake of understanding, the initial fluorescence is measured after the tissue is exposed to darkness for a certain period, which causes the oxidation of electrons, optimizing the absorption of light energy, i.e., this is the moment when there is the lowest loss by fluorescence, so the values are lower. However, if the photosystem is under stress, the values tend to increase, becoming an indicator of the stress condition [18,34–36], as observed in the present study.

On the other hand, the opposite occurs in the maximum fluorescence: the plant under stress conditions cannot reach the energy boost limit, and rootstocks such as TSKC × (LCR × TR)—040 and TSKC × CTARG—019 showed a condition of exhaustion of

the photosynthetic apparatus, which may result from an attempt to make greater use of the energy received.

The changes observed in minimum fluorescence and maximum fluorescence resulted in an effect on the variable fluorescence, which is obtained by their subtraction (Figures 8–10). However, the values of quantum efficiency of photosystem II were not affected by the interaction between the factors, which leads to the hypothesis that the photosynthetic apparatus did not suffer significant damage, i.e., despite being under stress conditions, the plants were able to maintain photosynthetic metabolism. Despite that, compared to the first year of cultivation, it was possible to observe a small reduction in the quantum efficiency of photosystem II, which changed from 0.78 in the first year to 0.75 in the second year (Table 8), a value within the normal range [37]. In similar studies with control plants, these authors reported the range of variation for the quantum efficiency of photosystem II between 0.72 and 0.81 as ideal for citrus plants.

### 4.4. Gas Exchange

The effect of salinity on gas exchange was significant in the first year of cultivation, although the values of net photosynthesis were within the range described for citrus plants, from 4 to 10 $\mu$mol $CO_2$ $m^{-2}$ $s^{-1}$ [15]. The values found can be considered normal, even with irrigation with water of higher salinity (4.8 dS $m^{-1}$), except for TSKC $\times$ CTTR—012 citrangedarin, whose value was below 4 $\mu$mol $CO_2$ $m^{-2}$ $s^{-1}$.

The gas exchange variables were more sensitive to the effect of salinity and allowed the selection of rootstocks with greater physiological adaptation capacity, especially through the evaluation of the relative reduction in stomatal conductance and transpiration of plants, as observed in 'Sunki Tropical' mandarin, which was classified among the best genotypes when irrigated with water of 4.8 dS $m^{-1}$, and the hybrids HTR—069 citrangor and TSKC $\times$ (LCR $\times$ TR)—040 citrimoniandarin, which did not suffer reductions when subjected to this salinity level, denoting the lower sensitivity of these individuals [8,33], the opposite of what occurred with TSKC $\times$ CTARG—019 citrangedarin, the most sensitive genotype to salinity.

On the other hand, it was possible to observe that most genotypes kept similar means, with no significant differences until the use of an ECw of 2.4 dS $m^{-1}$, and the values were reduced when the salinity level was increased to 4.8 dS $m^{-1}$, which points to the condition of stability or acclimatization since plants were in their second year of production and under irrigation with saline water. Furthermore, it should be noted that the mean values of net photosynthesis, at all salinity levels, were below 4.0 $\mu$mol $CO_2$ $m^{-2}$ $s^{-1}$, which can be interpreted as a condition of plant exhaustion, in the face of multiple stresses, notably due to the cultivation in the pot, because even if adequate nutritional management was applied, the restrictive conditions of the pots may have generated other stresses, such as those related to temperature [23,24], leading to a reduction in physiological potential.

The reduction caused by salinity in the $CO_2$ assimilation rate is reported in the literature as an osmotic effect [19] since the increase in the salt concentration reduces the water potential in the soil and reduces water absorption, causing stomata closure to ensure the maintenance of the plant's turgor, thereby reducing the $CO_2$ influx into the substomatal chamber.

The higher sensitivity to salinity observed in the hybrid TSKC $\times$ CTARG—019, based on physiological data, was accompanied by a lower fruit production by this rootstock. Although the application of water with an ECw of 4.8 dS $m^{-1}$ did not cause a significant loss of yield compared to plants irrigated with water of 0.14 dS $m^{-1}$ (Figure 4), there was a significant reduction in the production of plants grafted onto this citrangedarin.

This fact, however, only points to the importance of physiological variables in the understanding and indication of stress situations, which can be a very useful tool for decision-making, since the reduction caused by salinity was also observed in citrus plants under water salinity, with deleterious effects [37].

Considering that stomatal conductance reflects the process of $CO_2$ entry and water vapor exit for the performance of net photosynthesis, the results were coherent since the citrangedarin TSKC $\times$ CTARG—019 was the rootstock that also conferred the scion variety the lowest rates of net photosynthesis and stomatal conductance when it was irrigated with waters of higher salinity. In line with the results obtained in the present study, Sousa et al. [38] highlight that stomatal conductance is responsible for controlling water exit and $CO_2$ entry, which in turn is one of the main substrates for photosynthesis. Thus, as the mean of stomatal conductance is modified by the salinity level of irrigation water, the stomata begin to be hampered in their natural activities, compromising $CO_2$ assimilation.

## 5. Conclusions

Water salinity reduced the production of citrus plants, especially the number of fruits per plant (NFPL), the average fruit weight (AFW), and consequently the total weight of fruits (WFPL), and gradual reductions were observed in the two years of cultivation.

'Riverside' citrandarin, 'Sunki Tropical' mandarin, and HTR—069 citrangor showed greater production stability, even with the increase in irrigation water salinity.

Genotypes 'Tahiti' grafted on 'Sunki Tropical' mandarin, TSKC $\times$ (LCR $\times$ TR)—040 citrimoniandarin, and HTR—069 citrangor were more tolerant to an ECw of 4.8 dS m$^{-1}$.

The effect of salinity on citrus plants was osmotic, reducing their net photosynthesis, transpiration, and stomatal conductance.

The photosynthetic apparatus of the scion/rootstock combinations was not affected by salinity since the quantum efficiency of photosystem II (Fv/Fm) was higher than 0.78 in the first year of cultivation and equal to 0.75 during the second year of cultivation.

Citrangedarins TSKC $\times$ CTTR—012, TSKFL $\times$ CTTR—013, and TSKC $\times$ CTARG—019 citrangors showed higher sensitivity to salt stress, with the lowest yields when irrigated with saline water of 2.4 dS m$^{-1}$ and with the largest reductions in quantum efficiency of photosystem II (PSII) and gas exchange.

Irrigation water of up to an electrical conductivity of 2.4 dS m$^{-1}$ can be used in the cultivation of citrus plants without significantly compromising their physiology, provided that salt-tolerant rootstocks are used and a leaching fraction of 0.10 is applied.

**Author Contributions:** Writing—review and editing, G.O.M., M.E.B.B., H.R.G., A.S.d.M. and W.S.S.F.; visualization, R.R.G.F., A.S.d.M., P.D.F., H.R.G., W.S.S.F. and M.E.B.B.; supervision, A.S.d.M., P.D.F. and H.R.G.; Data curation, G.O.M., S.S.S., E.R.E., R.R.d.M.N. and M.E.B.B.; project administration, M.E.B.B.; funding acquisition, M.E.B.B. All authors have read and agreed to the published version of the manuscript.

**Funding:** National Council for Scientific and Technological Development: 406460/2018-3.

**Institutional Review Board Statement:** Not applicable.

**Data Availability Statement:** Not applicable.

**Conflicts of Interest:** The authors declare no conflict of interest.

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
