# Peer review of "Salt Tolerance Indicators in ‘Tahiti’ Acid Lime Grafted on 13 Rootstocks"

_agriculture, doi:10.3390/agriculture12101673_

Round 1

Reviewer 1 Report

The manuscript by Martins et al., “SALINITY TOLERANCE INDICATORS IN ‘TAHITI’ ACID LIME GRAFTED ON 13 ROOTSTOCKS” (agriculture-1928412), showed the many results by gas exchange, chlorophyll a fluorescence and an interesting research by rootstocks and salinity tolerance in Citrus yield and productions.

The authors have done a good work, with many relevance photosynthesis and fluorescence analyses which employing various references based on a scientific method and structure. The “big data” were analysed and proved our understanding of salinity stress tolerance. The introduction, M&M, the results and discussion topic is good and minor points its necessary by adjusting in text. This article, which demonstrated great and relevant results and discussion, but few recent literature was cited. However, the manuscript can be accept to Agriculture journal after minor revision.

#Please, observe and standardize the terms throughout in a manuscript (gas exchange and chlorophyll a fluorescence analysis).

Minor points:

-Title: only the first capital letter of each word;

-Alphabetic order keywords;

-L58-62: Rewrite. Very extensive; Check all paragraphs. Sometimes the idea becomes too extensive (also, in results and discussion topics)  and the focus is lost to readers.

-Not “dot” in table; check all manuscript;

-®; superscript;

L186; 40 minutes was sufficiently? Previous test to better time?

L196 (Ci; unit?);

-L195-196: Photosynthetic variables: A;E; gs, Ci, A/E, etc...

-L231- Why 0-0.20, but not 0.20-0.40? Root influences?

-Table 3: Why decreased pH in 1 unit, between ffirst ans second year? Its influence in data analysis? For example, nutrient availability, salt concentration, root development, transpiration, stomatal conductance, hormone mobilization? Could discuss this physiological and productive approach in your discussion.

-L245: g plant-1; g fruit-1; Not dots, check all manuscript;

-Figures 1-16; except (Fig. 7): In general, Why legends added in figures? Its not correct!

-Tables, chlorophyll a fluorescence: Very strange these Fv/Fm values in your tables. Regardless of the equipment (absolute or relative intensity), the Fv/Fm ratio cannot result in such low values. Example Table 7 (Gen) Fv/Fm= 0.835, but not 0.0007. In addition, Why mean squares?

L422: Fv and Fm 328638, I think a report error? An absolute values its not important but a relation Fv/Fm!! Check, please?!

-Chl a fluorescence data: Why not use the values obtained directly from the equipment? In addition, NPQ, qN, QP, qL,qI and qE, could they not have been reported?

-My question: In fact these data are indicative of the efficiency of photosystem II in the dark... However, what about under light?

-Tables in special photosynthetic parameters: I’m suggesting to author added the unit at variables;

-L595-601; extensive paragraphs;

L693-700; For example, How can a productor  to monitoring? Here, the authors showed no significant differences. Physiologically, how does this change at the PSII level occur, with the chlorophyll a fluorescence and gas exchange data analysed?

-References: As a suggestion, update to recent literature in manuscript and discute in your manuscript; Many references are more than 8 years old. In addition, on 30 pages (916 lines), only 38 references were cited.

Best Regards

Author Response

Dear reviewer, in answer to questions, we highlight:

#Please, observe and standardize the terms throughout in a manuscript (gas exchange and chlorophyll fluorescence analysis).

We dit it.

Minor points:

-Title: only the first capital letter of each word;

Ok

-Alphabetic order keywords;

Ok

-L58-62: Rewrite. Very extensive; Check all paragraphs. Sometimes the idea becomes too extensive (also, in results and discussion topics)  and the focus is lost to readers.

Ok, we adjust it.

-Not “dot” in table; check all manuscript;

Ok, we adjust it.

-®; superscript;

Ok, we adjust it.

L186; 40 minutes was sufficiently? Previous test to better time? Yes, we do these tests with citrus plants, and see that from 30 min sufficiently, but this data didn’t publicised.

Ok, we adjust it in text.

L196 (Ci; unit?);

Ok, we adjust it.

-L195-196: Photosynthetic variables: A;EgsCiA/E, etc...

Ok, we adjust it.

-L231- Why 0-0.20, but not 0.20-0.40? Root influences?

The 020-0.40 samples were collected, but, only data relative to 0.0-0.20 were showed due more root concentrations.

-Table 3: Why decreased pH in 1 unit, between first and second year? Its influence in data analysis? For example, nutrient availability, salt concentration, root development, transpiration, stomatal conductance, hormone mobilization? Could discuss this physiological and productive approach in your discussion. Discussions add.

Ok, we adjust it in results and discussion.

-L245: g plant-1; g fruit-1; Not dots, check all manuscript;

Ok, we adjust it.

-Figures 1-16; except (Fig. 7): In general, Why legends added in figures? Its not correct!

The legends were add to make the figures self-explanatory.

-Tables, chlorophyll a fluorescence: Very strange these Fv/Fm values in your tables. Regardless of the equipment (absolute or relative intensity), the Fv/Fm ratio cannot result in such low values. Example Table 7 (Gen) Fv/Fm= 0.835, but not 0.0007. In addition, Why mean squares?

Due this data are relative to summary of variance analysis

L422: Fv and Fm 328638, I think a report error? An absolute values its not important but a relation Fv/Fm!! Check, please?!

This data are relative to mean of square, from of the variance analysis.

-Chl a fluorescence data: Why not use the values obtained directly from the equipment?

This data can be viewed in Figures 8, 9, and 10, in the table we can see the Mean square from the summary of variance analysis.

In addition, NPQ, qN, QP, qL, qI and qE, could they not have been reported?

The fluorescence protocol used was OJIP, for these variables, it’s necessary to the Kinect protocol.

-My question: In fact, these data are indicative of the efficiency of photosystem II in the dark... However, what about under light?

The determination of the fluorescence in the dark, in fact, refers to the determination of the fluorescence from leaves after adaptation to the dark, when the chlorophylls are in a reduced condition, that is, with the full potential to the oxidation. Fluorescence under bright conditions can also be performed using other protocols, such as Yield or Kinect protocols, however, they were not addressed in this paper.

-Tables in special photosynthetic parameters: I’m suggesting to author added the unit at variables;

Ok, we did it.

-L595-601; extensive paragraphs;

Adjusted

L693-700; For example, How can a productor  to monitoring? Here, the authors showed no significant differences. Physiologically, how does this change at the PSII level occur, with the chlorophyll a fluorescence and gas exchange data analysed?

Adjusted

-References: As a suggestion, update to recent literature in the manuscript and discute in your manuscript; Many references are more than 8 years old. In addition, on 30 pages (916 lines), only 38 references were cited.

Ok

Reviewer 2 Report

I have gone through the manuscript “SALINITY TOLERANCE INDICATORS IN ‘TAHITI’ ACID LIME GRAFTED ON 13 ROOTSTOCKS”. The work done by the authors is of great concern related to citrus, which is one of the most significant fruit. Experiment performed and data presented is of significance in improving our information regarding citrus. Before the manuscript be accepted for publication, the authors should address the following concern:

Abstract:

Comments:

Title: Can be made it more attractive

Abstract: Please use ‘articles’ appropriately; for example, add ‘the’ as and when required. The abstract could be further elaborated in terms of shortening to much initial intro, adding more content in methodology, results and achievements and future implementation.

Keywords: Refined the key words

Introduction: This section is well written, however, needs a revision for proper English. Some more text in the introduction section can be added

Materials and Methods: Methods are well explained. Although it still requires English improvements

Additionally why paragraph has been made shorter

Results: I have critical comments regarding figures, it must be presented in different styles, why much text right side of the figures.

The discussion and conclusions are coherent with the results, clear and directly relevant to the research data.

References: Up to date and relevant, although the number of references can be increased. I can see most of the references are not properly formatted. Please revise the entire section.

Author Response

Dear reviewer, we do all adjust at the paper.

Title: Can be made it more attractive

Abstract: Please use ‘articles’ appropriately; for example, add ‘the’ as and when required. The abstract could be further elaborated in terms of shortening to much initial intro, adding more content in methodology, results and achievements and future implementation. Ok

Keywords: Refined the key words, OK

Introduction: This section is well written, however, needs a revision for proper English. Some more text in the introduction section can be added. Ok

Materials and Methods: Methods are well explained. Although it still requires English improvements. Ok

Additionally why paragraph has been made shorter. We do it to improve the explications.

Results: I have critical comments regarding figures, it must be presented in different styles, why much text right side of the figures.

The discussion and conclusions are coherent with the results, clear and directly relevant to the research data. Ok

References: Up to date and relevant, although the number of references can be increased. I can see most of the references are not properly formatted. Please revise the entire section. Ok

Best regards

Reviewer 3 Report

The aim was to study the salt content in the soil and physiological aspects of combinations of citrus rootstocks with saline water in irrigation during the first two years of cultivation.

The study was carried out in the experimental farm but detailed meteorological and amount of irrigation (per pots or/and per day) during the period of the experiment are missing in the material and methods section? Based on what the quantity of water was calculated and how it was applied to the plants? It is not detailed how water deficit irrigation was applied the irrigation schedule is not detailed. It mandatory to mention the field capacities used

Overall, the manuscript is well organized and well written. The title is informative and relevant. However, the references are not relevant and recent. Only one or two citations are 2022, the rest are not recent

The results presentation Is appropriate

The introduction was very informative and appropriately explaining the general importance, hypotheses and the aims.

The material and methods section sound well-presented and organized/ however the authors are invited to add more details on the field experimental conditions (full climatic data of the 2 years (ET0, radiation wind speed, lightening hours), irrigation methods (crop water requirement) and sampling method.

The discussion section is well presented however I suggest to also discuss why there is no effect of salinity on chlorophyll a fluorescence

Overall, the study is well organized and responds to the major aims designed in the introduction section with solid results. Relevant suggestions for future studies in the field are missed in the conclusion or the end of the discussion

Author Response

All suggestions were added in text.

Thanks for your contributions!

Best Regards